

# A sensitivity study on radiative effects due to the parameterization of dust optical properties in models

Ilias Fountoulakis[1,2], Alexandra Tsekeri[1], Stelios Kazadzis[3], Vassilis Amiridis[1], Angelos Nersesian[4], Maria Tsichla[1,5], Emmanouil Proestakis[1], Antonis Gkikas[1,2], Kyriakoula Papachristopoulou[1,6], Vasileios Barlakas[7,8], Claudia Emde[9], Bernhard Mayer[9]

[1] Institute for Astronomy, Astrophysics, Space Applications and Remote Sensing, National Observatory of Athens, Athens, Greece
[2] Research Centre for Atmospheric Physics and Climatology, Academy of Athens, Athens, Greece
[3] Physics and Meteorological Observatory Davos, World Radiation Center, Davos, Switzerland
[4] Sterrenkundig Observatorium Universiteit Gent, Belgium
[5] Environmental Chemical Processes Laboratory, Department of Chemistry, University of Crete, Greece
[6] Laboratory of Climatology and Atmospheric Environment, Department of Geology and Geoenvironment, National and Kapodistrian University of Athens, Athens, Greece
[7] EUMETSAT, Darmstadt, Germany
[8] HE Space Operations GmbH, Darmstadt, Germany
[9] Meteorological Institute, Ludwig-Maximilians-University, Theresienstr. 37, 80333 Munich, Germany

*Correspondence to*: I. Fountoulakis (ifountoulakis@Academyofathens.gr)

**Abstract.** Most of the dust models underestimate the load of the large dust particles, consider spherical shapes instead of irregular ones, and have to deal with a wide range of dust refractive index (RI) to be used. This leads to an incomplete assessment of the dust radiative effects and dust-related impacts on climate and weather. The current work aims to provide an assessment, through a sensitivity study, of the limitations of models to calculate the dust direct radiative effect (DRE) due to the underrepresentation of its size, RI and shape. We show that the main limitations stem from the size and RI, while the shape plays only a minor role, with our results agreeing with recent findings in the literature. At the top of the atmosphere (TOA) close to dust sources, the underestimation of size issues an underestimation of the direct warming effect of dust of ~18 - 25 W/m$^2$, for dust aerosol optical depth (DOD) of 1 at 0.5 μm, depending on the solar zenith angle (SZA) and RI. The underestimation of the dust size in models is less above the ocean than above dust sources, resulting in an underestimation of the direct cooling effect of dust above the ocean by up to 3 W/m$^2$, for AOD of 1 at 0.5 μm. We also show that the RI of dust may change its DRE by 80 W/m$^2$ above the dust sources, and by 50 W/m$^2$ at downwind oceanic areas, for AOD of 1 at 0.5 μm at TOA. These results indicate the necessity of including more realistic sizes and RIs for dust particles in dust models, in order to derive better estimations of the dust DRE, especially near the dust sources and mostly for studies dealing with local radiation effects of dust aerosols.



## 1 Introduction

One of the most abundant aerosols in the Earth's atmosphere is mineral dust with ~57% of its total load originating from North Africa (Huneeus et al., 2011; IPCC, 2013). Saharan dust has a significant impact on global climate through long-range-transported dust particles mainly over the Atlantic Ocean to America (Parkin et al., 1972; Doherty et al., 2008; Prospero et al., 2010) and over Europe and the Mediterranean Basin (e.g., Meloni et al., 2018; Papachristopoulou et al., 2022).The radiative forcing of dust on the global climate includes the direct interaction of the particles with the radiation in the atmosphere, i.e., through absorbing and/or scattering of the shortwave (SW) and longwave (LW) radiation, called the direct radiative effect (DRE), and through the indirect (IRE) and semi-direct (SRE) radiative effects of dust, through altering the physical and optical properties of clouds, and subsequently their radiative forcing. The quantification of both effects is currently uncertain (e.g., Adebiyi et al., 2022).

Whether desert dust aerosols warm or cool the planet by their DRE is still a matter of debate, with e.g. Kok et al. (2022) and Kok et al. (2023) reporting a DRE of -0.2 ± 0.5 Wm$^{-2}$. Mineral particles scatter and absorb SW radiation, causing cooling and warming, correspondingly, whereas induce a warming through scattering and absorption of the LW radiation emitted by the Earth's surface. Dust particles of radius smaller than 1 μm tend to cause cooling in the SW, since scattering dominates over spectral absorption in the SW, while for larger particles the warming due to absorption in the SW is not negligible (e.g., Song et al., 2022). Moreover, in the LW, the effect of absorption becomes comparable to scattering, both for fine and coarse particles (e.g., Song et al., 2022). Dust models typically exclude the larger dust particles and underestimate coarse-mode concentrations, introducing uncertainties on the assessment of dust-induced impacts on climate, weather and biochemistry (e.g., Kok et al., 2022, Adebiyi and Kok, 2022; Drakaki et al., 2022).

Regarding dust IRE, through altering the properties of clouds larger particles may produce cloud condensation nuclei, and initiate precipitation depending on their mixture with soluble material (e.g., Adebiyi et al., 2022). Larger dust particles are also more effective ice condensation nuclei (e.g., Petters and Kreidenweis, 2007; Diehl et al., 2014) than smaller dust particles which are less hygroscopic (e.g., Ibrahim et al., 2018). Thus, the dust size has an impact on the amount, properties and spatial distribution of clouds, and subsequently on global precipitation and climate (e.g., Nenes et al., 2014; Karydis et al., 2017). Larger dust particles also contribute more to dust mass, which controls the impact of dust on ocean and tropical rainforest ecosystems (Jickells et al., 2005; Yu et al., 2015), and also on the ocean carbon cycle (van der Does et al., 2018).

In more detail, the radiative effects of dust particles depend on their microphysical properties, i.e., the size, shape, orientation, and refractive index (RI) (e.g., Ulanowski et al., 2007; Haywood et al., 2001, 2003). Regarding the size, even state-of-the-art dust models heavily underestimate the portion of the dust burden attributed to large mineral particles, considering a maximum radii of 10 $\mu m$ (e.g., Mahowald et al., 2014; Adebiyi and Kok, 2022), while in the widely-used OPAC scattering database the maximum radius of coarse mode aerosols is 30 $\mu m$ (Koepke et al., 2015). The size of the coarser dust grains is found to be much larger, with radii of up to >200 $\mu m$ (e.g., Ryder et al., 2019; van der Does et al., 2018; Kandler et al., 2011). Moreover, a significant fraction of particles with radii between 10 and 50 μm have been measured in many experimental campaigns in



North Africa (e.g., Weinzierl et al., 2009). Ryder et al., (2019) reported that excluding the larger particles over the Sahara desert can lead to an underestimation of both shortwave and longwave extinction by up to 18% and 26% respectively. Kok et al., (2017) found that by using an observationally constrained particle size distribution (SD) of dust at emission, global model calculations of dust radiative forcing resulted in less cooling compared to previous estimates from AeroCom models (Huneeus et al., 2011), wherein smaller, more cooling particles were over-represented and coarser, more warming particles were underestimated.

In addition, many dust models consider a spherical shape for dust grains, although the grains have irregular shapes (e.g., Kandler et al., 2009). The calculations of the optical properties are much simpler for spherical particles, but their deviation from the optical properties of dust with non-spherical shapes has been shown to affect the quantification of the dust radiative effect. For example, Otto et al. (2011) have quantified the effect on dust direct radiative forcing when considering spheroids, to be 5% at TOA over land, 55% at TOA over ocean, and 15% at the bottom of the atmosphere (BOA) over land and ocean. Ito et al. (2021), considering ellipsoidal shapes instead of spheres, reported no effect at TOA, and an atmospheric warming of 0.18 W/m$^2$ at BOA. Irregular hexahedral shapes have also been used to describe the optical properties of dust (Saito et al., 2021), but not for estimating its radiative effects. More realistic irregular shapes have been also suggested (Gasteiger et al., 2011), but not for a wide range of sizes and RIs, due to their costly computations.

Moreover, modelling studies commonly use RIs (e.g. Patterson et al., 1977; D'Almeida et al., 1991, Shettle and Fenn, 1979; Sokolik et al., 1993) that result in systematic overprediction of the absorption of mineral dust compared to satellite retrievals (Balkanski et al., 2007). RIs with lower imaginary parts (e.g., Balkanski et al., 2007; Colarco et al., 2014) are considered to be more appropriate and are consistent with recent observations for Saharan dust (Rocha-Lima et al., 2018). Song et al. (2022) have provided a quantification of the DRE on a global scale for different sizes, RIs and shapes (spheres vs spheroids) for dust. According to their findings, the RI used in aa model is the most important factor in determining the dust DRE, inducing large uncertainties, up to 45%. The authors also showed that not considering very large particles results in uncertainties of 15–20 % in dust DRE at TOA and in the atmosphere, while dust non-sphericity induces a negligible effect.

Other important parameters for a complete assessment of the dust radiative effects are the surface albedo, the dust layer height (e.g., Liao, H. and Seinfeld, 1998) and the amount of water vapour found in dust layers (Gutleben et al., 2019; 2020; Ryder, 2021). Regarding the surface albedo, the induced SW cooling is enhanced above dark surfaces (e.g., the ocean), since the presence of dust hinders the absorption of the incoming radiation at the surface. On the contrary, SW warming is enhanced above bright surfaces (e.g., desert) because in the absence of dust the surface would reflect more incoming radiation (Kok et al., 2022 and references therein).

Herein a sensitivity study has been conducted emphasizing on the determinant factors for the quantification of dust DREs in models, such as mineral particles' size, shape and composition. More specifically, we investigated the effects of:

- Neglecting particles with diameters larger than 10 μm in the simulations, as done in dust models
- Assuming spheroidal instead of the spherical particle shapes consideredconsidered in dust models



- Taking into account accountthe wide range of RIRI of dust particles (representing different particle compositions), not considered in dust models

Moreover, the dust-induced radiative SW and LW effects, both at BOA and TOA under clear-sky conditions, are quantified above desert and ocean, for varying dust loads and different sun geometries. It is important to clarify at this point that our study
105  aims to contribute towards the understanding of how the dust optical properties parameterization in models affect the calculated dust DRE under different atmospheric and land surface conditions, rather than providing quantitative estimates of the (local or global) dust radiative effects. Small-scale measurement-based modelling studies are necessary to understand the interactions between dust and solar/thermal radiation, which in turn, would improve the accuracy of the large-scale modelling simulations (Yu et al., 2006). Section 2 presents the data and methodology followed in the study, Sect. 3 presents the results, Sect. 4
110  provides a summary and comparison with other studies, and Sect. 5 presents the main findings of this study.

## 2 Data and methodology

We quantified the effect on the radiation fluxes and DRE of dust due to the underestimation in size and the assumption of sphericity in models, using more realistic shapes, SDs and RIs for the dust particles. Optical properties were calculated using the MOPSMAP scattering database (Modelled optical properties of ensembles of aerosol particles; Gasteiger and Wiegner,
115  2018), considering spherical or spheroidal shapes for the dust particles. The radiative transfer calculations were performed using the radiative transfer solver MYSTIC (Monte carlo code for the phYSically correct Tracing of photons In Cloudy atmospheres; Mayer, 2009).

### 2.1 Microphysical properties of dust particles

*Volume size distributions*
120  The volume size distributions used were acquired from airborne in-situ measurements above dust sources in the Sahara desert, and in the Saharan Air Layer (SAL) above the ocean, during the Fennec experimental campaign (Ryder et al., 2019). Fennec was held in June 2011, above the Sahara Desert and above the ocean close to the Canary Islands (Fig. 1). Fennec campaign was selected in this study since it provides size distributions near to dust sources, with radii as large as ~150 μm above the sources. A comparison with respect to the findings of other experimental campaigns is shown in Fig. 9 in Ryder et al. (2019).

125



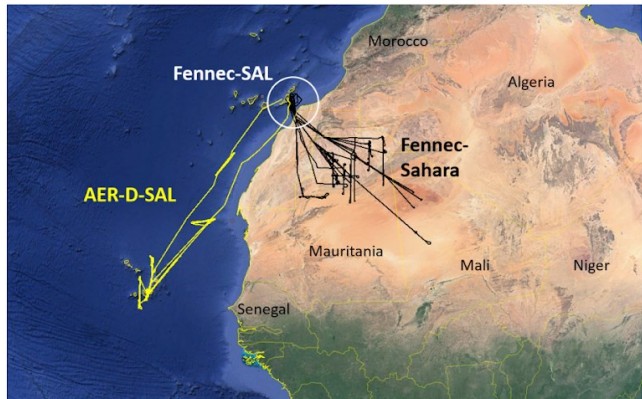

**Figure 1:** The location of the Fennec campaign, with measurements above dust sources in Sahara desert ("Fennec-Sahara" -black lines), and above the ocean in the SAL ("Fennec-SAL" -white circle). The measurements from the AER-D campaign (yellow lines) were not used herein. Reproduced from Ryder et al. (2019) under CC-BY license.

Figure 2 shows the mean log-fit of the volume size distributions measured for dust particles during the Fennec campaign, above dust sources and above the ocean in the SAL. A detailed presentation about the in-situ measurements can be found in Ryder et al. (2013a), (2013b) and (2019). Equation 1 provides the formulation of the measured volume size distributions, as the summation of four log-normal modes, and Table 1 provides the corresponding parameters for each mode, above desert and ocean (Ryder et al. 2013b; 2019).

$$\frac{dV}{dlog(r)} = \sum_{i=1}^{i=4} \left[ \frac{N_i}{\sqrt{2\pi} log(\sigma_i)} exp\left( -\frac{1}{2} \left( \frac{log(r) - log(r_{mod,i})}{log(\sigma_i)} \right)^2 \right) \frac{4\pi r^3}{3} \right] \qquad \text{Eq.1}$$

$N_i$ is the total number of particles ($\# \ cm^{-3}$) of log-normal mode $i$, $r$ is the particle radius, $r_{mod,i}$ is the median radius of log-normal mode $i$ (with respect to the number size distribution), and $\sigma_i$ is the width of log-normal mode $i$.

In many models the maximum radius of dust particles spans up to 10 μm (since dust is not spherical, the "radius" usually refers to the radius of the volume-equivalent sphere), whereas the measurements have shown that particles with radii even more than 50 μm can be found. The size distributions of the dust particles considered in the models are called herein SD10 (i.e., with maximum radii of 10 μm), whereas the more realistic size distributions of dust are called SD50 (i.e., with maximum radii of 50 μm). The size distributions of SD10 and SD50 above desert and ocean are shown in Fig. 2.





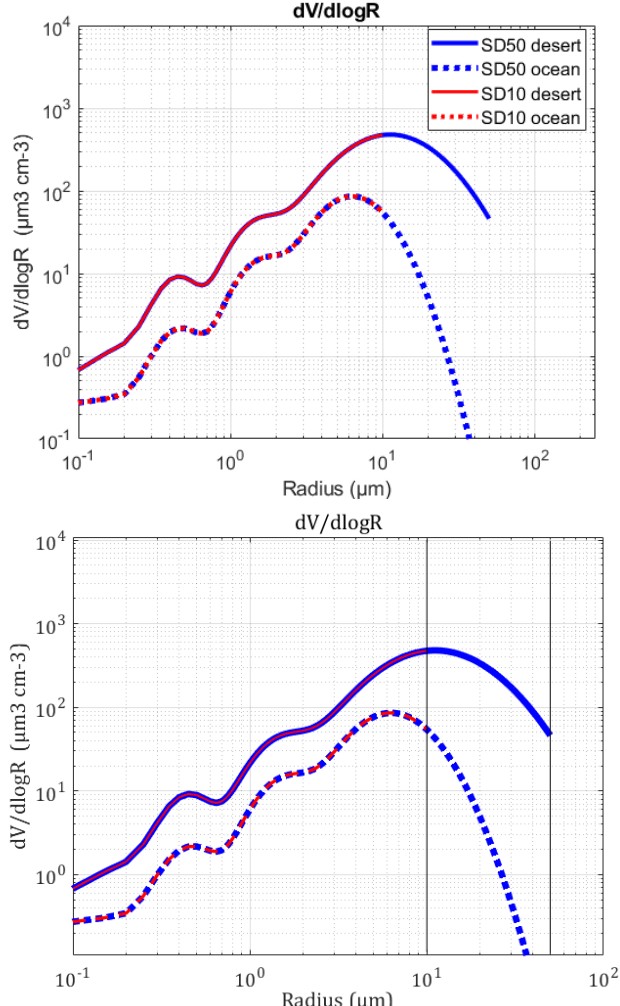

**Figure 2:** Mean log-fit of the volume size distributions measured during the Fennec campaign above the desert (solid lines) and above the ocean in the SAL (dash lines). The volume size distributions of "SD50" with maximum radius up to 50μm, are denoted with blue. The volume size distributions of "SD10" with maximum radius up to 10μm, are denoted with red.

**Table 1:** The parameters of the mean log-fit volume size distributions for the volume size distributions above desert and ocean in Fig. 2.

| | Desert | | | Ocean | | |
|---|---|---|---|---|---|---|
| Mode # | $N$ | $r_{mod,i}$ | $\sigma$ | $N$ | $r_{mod,i}$ | $\sigma$ |
| 1 | 508.27 | 0.025 | 2.5 | 946.031 | 0.012 | 2.5 |
| 2 | 8.84 | 0.355 | 1.33 | 2.599 | 0.3435 | 1.38 |
| 3 | 1.89 | 1.02 | 1.45 | 0.559 | 1.087 | 1.438 |
| 4 | 0.54 | 2.64 | 2 | 0.122 | 3.1385 | 1.62 |



*Refractive indices*

We used three different spectral-resolved RIs for dust particles, as these are provided from OPAC scattering database (d'Almeida et al., 1991; Hess et al., 1998), the World Meteorological Organization (WMO, 1983), and Balkanski et al. (2007) (Fig. 3). As shown in Ryder et al. (2019), these refractive indices represent the maximum, mean and minimum values for the imaginary part of the refractive index of dust particles in the visible and near-infrared, as these are reported in the literature, thus covering the whole range of the reported values for dust.


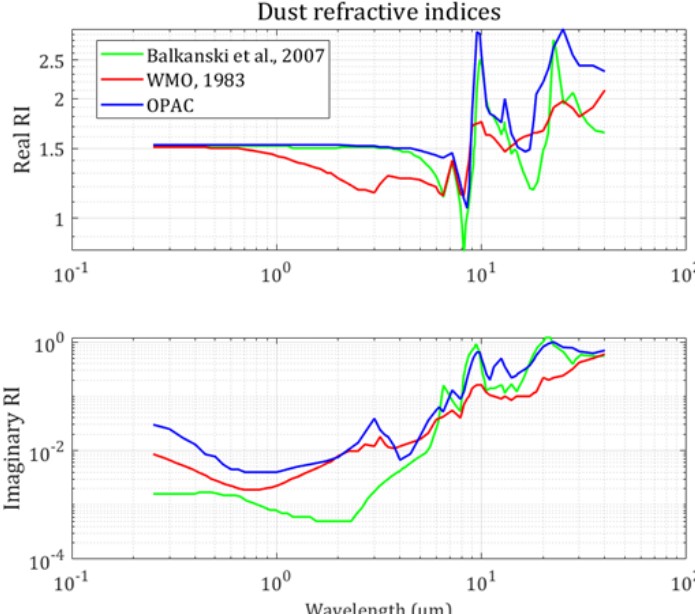

**Figure 3:** Spectrally resolved real (upper panel) and imaginary (bottom panel) parts of the complex RI of desert dust particles, derived from Balkanski et al. (2007) (green), WMO (1983) (red), and OPAC (blue; d'Almeida et al., 1991; Hess et al., 1998).


***Dust particle shape***

The dust particles are typically irregularly shaped, thus their optical properties are not easy to obtain for a wide range of RIs and wavelengths, especially for the larger sizes (e.g. Gasteiger et al., 2011). Herein, we use spheres and spheroids to model the shape of dust particles, with the latter being the widely-used alternative for modelling the non-spherical shape of dust grains

(e.g. Dubovik et al., 2006). The aspect ratio distribution of the spheroids was measured during the Saharan Mineral Dust Experiment (SAMUM) campaign in southern Morocco (Kandler et al., 2009), and is shown in Fig. 4 (the aspect ratio distribution is normalized by the total number of particles).





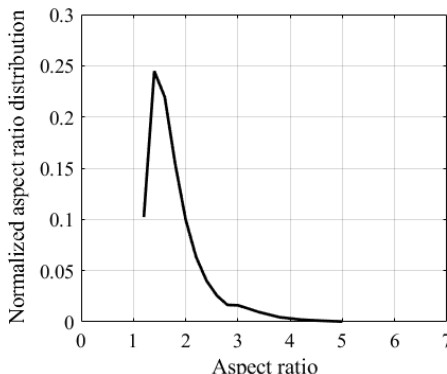

 **Figure 4:** The aspect ratio distribution of the spheroids used to model the non-spherical shape of dust in the current study, normalized by the total number of particles. The aspect ratio distribution was measured during the SAMUM campaign in southern Morocco (adapted by Kandler et al., 2009).

## 2.2 Optical properties

We calculated the optical properties of SD10 and SD50 dust particles using the MOPSMAP scattering database (Gasteiger and

Wiegner, 2018). In MOPSMAP the optical properties are pre-calculated for single particles, for a wide range of sizes and RIs, for spheres, spheroids, and a set of irregular particle shapes (the latter for a small range of sizes and RIs, thus they cannot be used here). The scattering properties of the spherical particles are pre-calculated in MOPSMAP using the Mie theory (Mishchenko et al., 2002; Mie, 1908). For the spheroidal particles the scattering properties are derived with the T-matrix method (TMM) (Mishchenko and Travis, 1998) for size parameters (i.e., $\frac{2\pi r}{\lambda}$, where $r$ is the radius of the particle and $\lambda$ is the

wavelength of light) 5-125, and the improved geometric optics method (IGOM) by Yang et al. (2007) and Bi et al. (2009) for size parameters >125. The optical properties of the particle ensembles are derived after interpolating the discrete values and performing ensemble averaging (Gasteiger and Wiegner, 2018).

The optical properties that are used as inputs for the radiative transfer calculations are the extinction coefficient, the single scattering albedo (SSA), and the phase function, all for a spectral range of 0.35 - 40 μm. For the temperatures at the Earth's

surface considered in this study, 4 - 5% of the total longwave radiation is emitted in the range 40 - 150 μm. As in other similar studies (e.g., Ryder et al., 2019) we consider only wavelengths that are shorter than 40 μm, mainly because RIs are not available for longer wavelengths. Nevertheless, changes in aerosol properties do not practically impact the transfer of terrestrial radiation at wavelengths longer than 40 μm in the atmosphere, since water vapor absorbs all radiation in the particular spectral region. Figure 5 shows the extinction, scattering and absorption coefficients for SD10 (dashed lines) and SD50 particles (solid lines),

above desert and ocean. For these calculations we consider the volume size distributions discussed in Section 2.1 (Fig. 2). Moreover, the optical properties are calculated for the different RIs of dust used herein, provided by Balkanski et al. (2007) (green line), WMO (1983) (red line), and OPAC scattering database (blue line).





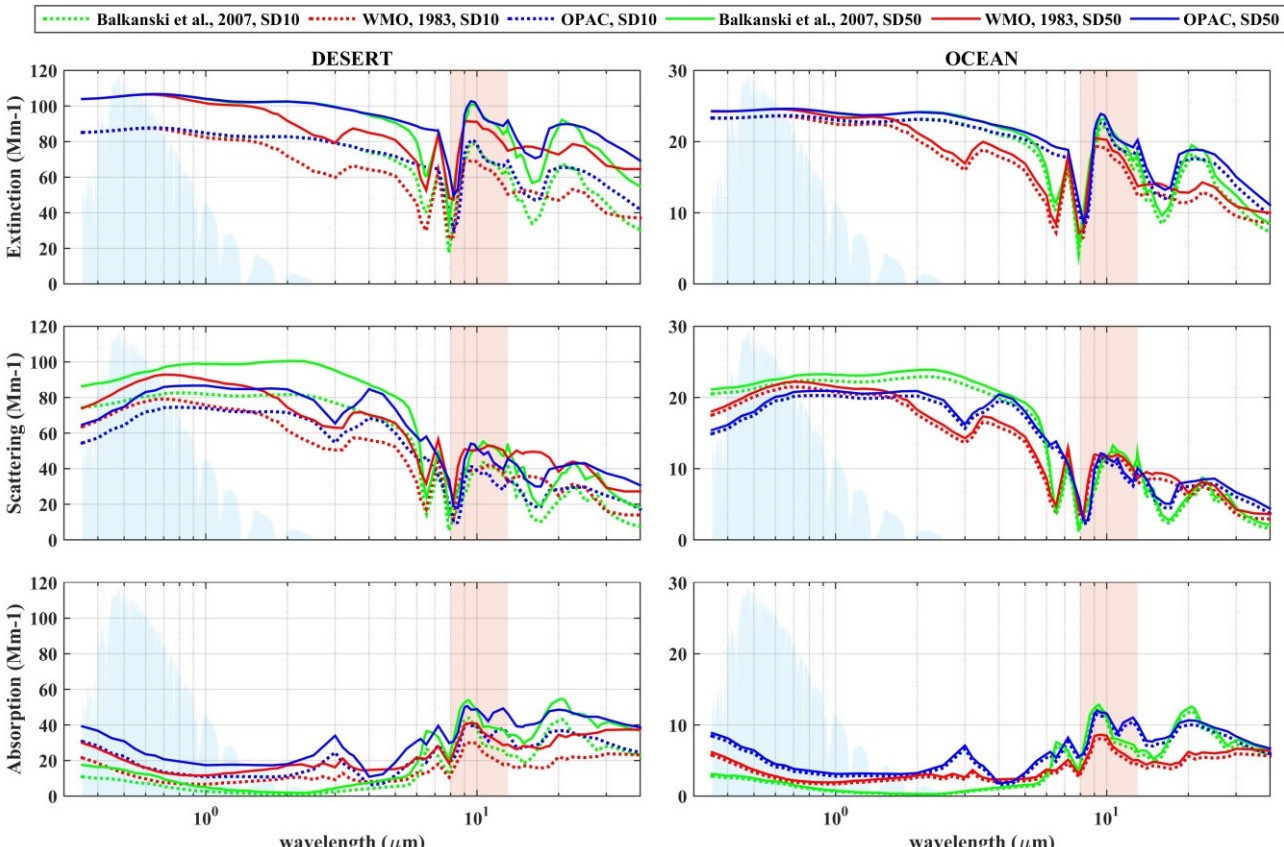

**Figure 5:** Spectral dependence of the extinction (upper panels), scattering (middle panels), and absorption coefficients (bottom panels), for SD10 (dashed lines) and SD50 dust particles (solid lines), above desert (left) and ocean (right), for the three different dust RIs from Balkanski et al. (2007) (green), WMO (1983) (red), and OPAC (blue). The light blue shadowed area represents the normalized spectrum of the SW irradiance (aerosol and cloud free atmosphere, SZA=0), while the shadowed red area represents the atmospheric window in the LW.

Above the desert (Fig. 5, left) considering the large (diameter > 10 μm) particles increases the scattering and absorption of solar and terrestrial radiation relative to the SD10 particles. Over the ocean the differences between SD10 and SD50 particles are much smaller (Fig. 5, right) mainly because there are less particles with diameters between 10 and 50 μm in the mixture. In general, scattering plays a more important role relative to absorption in the SW, whereas in the LW the scattering and absorption efficiencies are comparable (for both, SD10 and SD50).

It is a common practice in dust transport models to change the calculated dust mass concentration in order to reproduce aerosol optical depth (AOD) observations at specific wavelengths from ground or space (e.g., Drakaki et al., 2022). Thus, the same dust optical depth (DOD) (at 500 mn) has been used for the radiative transfer calculations (see Section 3), for the different SDs, shapes, and RIs considered in this study.

Considering the same DOD value at 0.5 μm means that the extinction coefficient is normalized at this wavelength (i.e., the overall aerosol mass and volume are implicitly modified in order to achieve the same DOD at 0.5 μm for different particles). Thus, at 0.5 μm, differences in the radiative effects of dust depend on the relative contribution of scattering and absorption





(i.e., the SSA). For different SDs and RIs, absorption and scattering coefficients are changing differently with wavelength, which means that the DOD does not remain the same for the different dust species with changing wavelength, and that differences in the SSA also depend on wavelength. To better understand how changing dust optical properties affect radiative transfer, the extinction, scattering and absorption coefficients normalized by their respective values at 0.5 μm, along with the SSA are presented in Fig. 6. As shown in Fig. 6b, d, f, h, the very small fraction of large particles in the aerosol mixture above

the ocean results in similar behaviour for SD10 and SD50. Over desert however, the fraction of large particles is higher considering SD50 instead of SD10, resulting in larger differences in scattering and absorption (Fig. 6a, c, e, g). Thus, for example, the same DOD at 0.5 μm results in higher AOD at 10 μm for SD50 relative to SD10. In the SW, the SSA (Fig. 6g) for SD50 is lower than the SSA for SD10 (i.e., the relative contribution of absorption is more important). In the region 8 – 13 μm (atmospheric window), as well as in the SW, differences in the SSA depend strongly on the used RI.


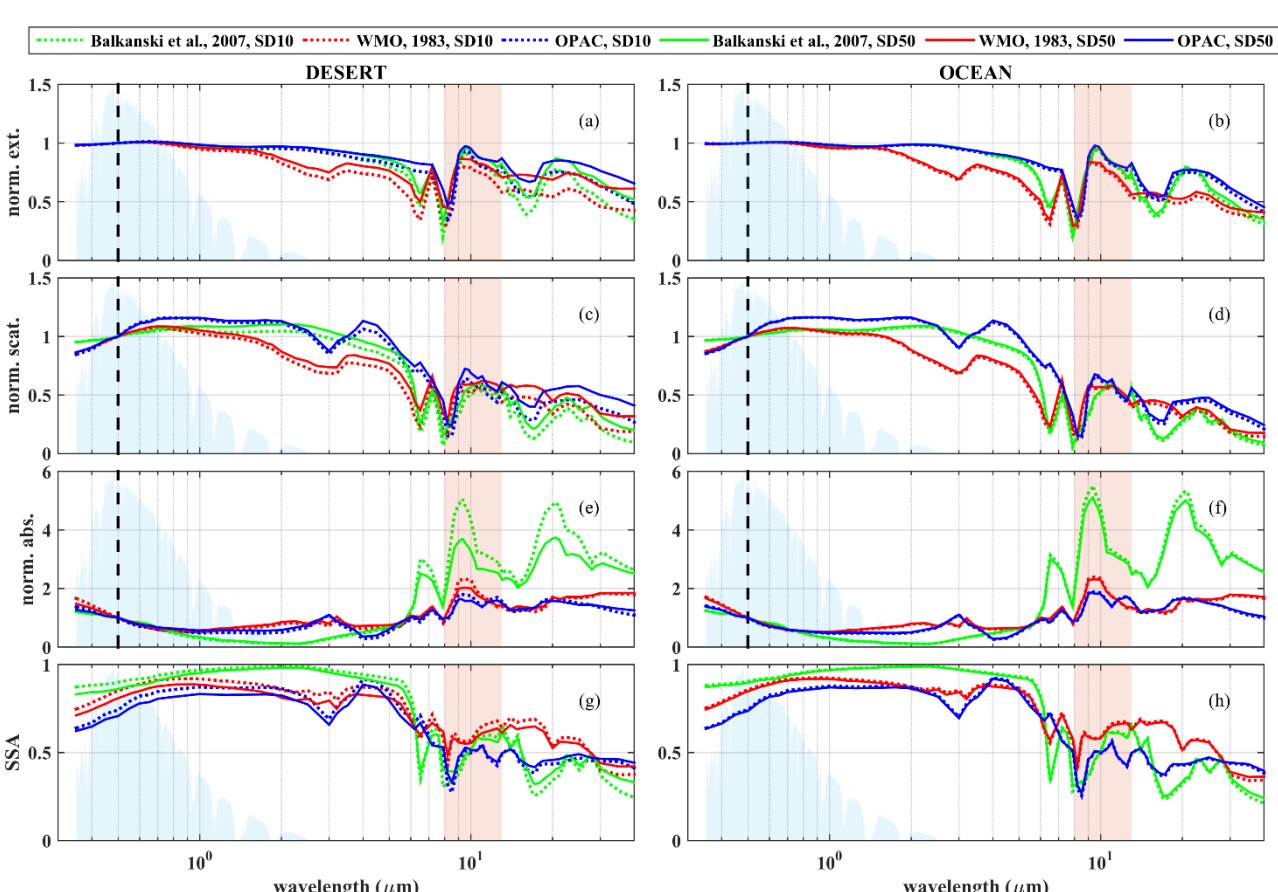

**Figure 6:** Spectral dependence of the normalized extinction coefficient (a, b), normalized scattering coefficient (c, d), absorption coefficients (e, f), and SSA (g, h), for SD10 (dash lines) and SD50 dust particles (solid lines), above desert (left) and ocean (right), for the three different dust RIs from Balkanski et al. (2007) (green), WMO (1983) (red), and OPAC (blue). Normalization has been performed relative to the
extinction, scattering, and absorption values at 0.5 μm. The light blue shadowed area represents the normalized spectrum of the SW irradiance (aerosol- and cloud-free atmosphere, SZA=0), while the shadowed red area represents the atmospheric window in the LW.



Figure 7 shows the effect of the shape in the optical properties of dust, when considering spheroidal shapes instead of spheres, for SD50 dust particles near the dust sources (blue solid line in Fig. 2). The differences for extinction, scattering and absorption coefficients are small, at 1-10% for the whole range of the spectrum, and for the three different RIs considered herein (Fig.

7a). The differences are even smaller for dust particles above the ocean (not shown here), due to the presence of fewer large dust particles. The differences are larger for the angular distribution of the scattered light, at large scattering angles, as shown for the normalized phase function at 0.5 μm in Fig. 7b. As discussed in Sect. 3.1, these differences do not result in large differences in the irradiance at TOA and BOA. This does not necessarily mean that the shape of dust does not play a role in the calculation of its optical and radiative properties. It only indicates that for spherical and spheroidal particles with the aspect

ratio distribution considered here (Fig. 4), the differences are small. This may not be the case though for more realistic shapes of dust particles (e.g. Saito et al., 2021; Gasteiger et al., 2011).

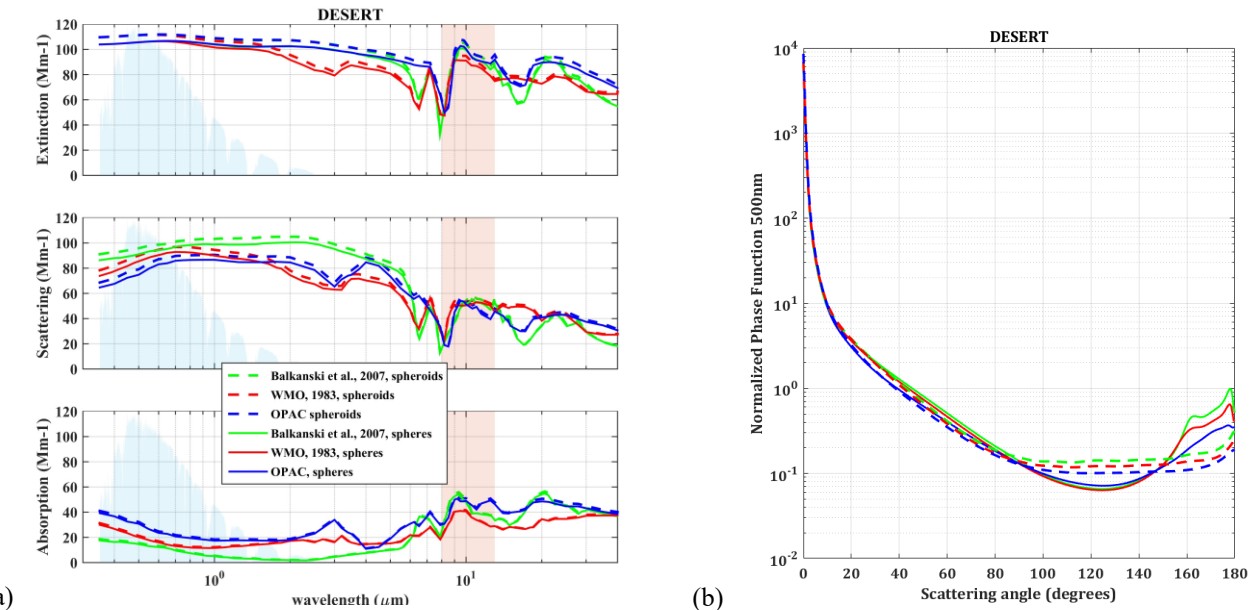

**Figure 7:** a) Spectral dependence of the extinction (up), scattering (middle) and absorption coefficients (bottom). The light blue shadowed area represents the normalized spectrum of the SW irradiance (aerosol- and cloud-free atmosphere, SZA=0), while the shadowed red area represents the atmospheric window. b) Normalized phase function at 0.5 μm. The optical properties are calculated for SD50 dust particles,

with spherical (solid lines) and spheroidal shapes (dash lines) above desert, for the three different dust RIs from Balkanski et al. (2007) (green), WMO (1983) (red), and OPAC (blue).

## 2.3 Radiative transfer modelling

Radiative transfer calculations were performed with libRadtran software (Mayer and Kylling, 2005; Emde et al., 2016).

LibRadtran is a library of radiative transfer solvers, atmospheric and topographic data, as well as various aerosol optical properties. Herein, we employed the radiative transfer solver MYSTIC (Mayer, 2009) which simulates the random path of the photons through the atmosphere (in one, two, or three dimensions in space) and calculates the radiance, irradiance and actinic





flux at any height. MYSTIC follows the Monte Carlo approach to determine a step width for each photon, the probability of the photon to be absorbed or scattered, and the new direction of the photon in case of scattering. MYSTIC calculations can be

performed in scalar mode neglecting polarization, and in vector mode where polarization is fully considered. The implementation of vector radiative transfer calculations in MYSTIC is described in detail in Emde et al., (2010). Using highly-peaked phase matrices increases computational time and the level of noise in the results. The amount of photons needed to yield a noise-free result can be diminished by applying variance reduction techniques. A sophisticated variance reduction technique is applied in MYSTIC, which works in the same way for scalar and vector simulations (Buras and Mayer, 2011).

Furthermore, MYSTIC applies periodic boundary conditions in x- and y- direction, which means that photons that leave the domain on one side at a certain vertical position, re-enter the domain on the opposite side at the same vertical position without changing their propagation direction.

We used MYSTIC and performed radiative transfer calculations in vector mode. MYSTIC was chosen instead of one of the simpler and faster (deterministic) solvers included in libRadtran (e.g., DISORT; Stamnes et al., 1988) because it allows to

include polarization in the simulations. Nevertheless, polarization was found to have very small effect on the results of the study. After a comprehensive testing, the number of photons selected for the simulations is $10^8$ photons, which results in differences that are well below 1% when the simulated spectral irradiance is of the order of 0.1 W/m$^2$/nm or larger. For weaker signals the uncertainty increases. However, spectral regions where the signal is weaker have a negligible impact on the calculation of the integrals of the SW and LW irradiances, and thus on the results of this study. The libRadtran settings that

were used for the study are summarized in Table 1.

**Table 1.** LibRadtran settings used for the radiative transfer simulations.

| Parameter | SW | LW |
|---|---|---|
| Atmospheric profile | AFGL tropical (Anderson et al., 1986) | AFGL tropical (Anderson et al., 1986) |
| Extra-terrestrial solar spectrum | Kurucz, 1994 | - |
| Molecular absorption parameterization | Representative wavelength approach (REPTRAN) (Gasteiger et al., 2014; Buehler et al., 2010) | Representative wavelength approach (REPTRAN) (Gasteiger et al., 2014; Buehler et al., 2010) |
| Solver | MYSTIC | MYSTIC |
| Wavelength range | 0.35 – 2.5 μm | 2.5 – 40 μm |
| SZA | 0°, 30°, 60° | 0°, 30°, 60° |
| CO2 mixing ratio | 420 ppb | 420 ppb |
| Number of photons | $10^8$ | $10^8$ |
| Polarization | Yes | Yes |
| Surface albedo | International Geosphere Biosphere | International Geosphere Biosphere |




|  | Programme (IGBP) (Loveland and Belward, 1997) | Programme (IGBP) (Loveland and Belward, 1997) |
|---|---|---|
| Surface albedo type | Bidirectional Reflectance Distribution Function (BRDF) | Bidirectional Reflectance Distribution Function (BRDF) |
| Total column of water vapor | 10 mm above desert, 30 mm above the ocean | 10 mm above desert, 30 mm above the ocean |
| Aerosol extinction coefficient profile | LIVAS (Proestakis et al., 2018) | LIVAS (Proestakis et al., 2018) |
| Columnar aerosol optical properties | Derived from MOPSMAP (phase matrixes) | Derived from MOPSMAP (phase matrixes) |
| DOD at 0.5 μm | 0.4, 1, 1.6 | 0.4, 1, 1.6 |
| Surface temperature | - | 323K for desert, 295K for ocean |

The spectral SW and LW irradiances at BOA and TOA were simulated assuming two different types of reflective surfaces:
ocean and desert. The total column water vapour (TCWV) was considered at 10 mm and 30 mm over desert and ocean, respectively, based on reanalysis data from the Modern-Era Retrospective analysis for Research and Applications, Version 2 (MERRA-2; GMAO, 2015) for the area of the study. Surface temperatures of 323 K and 295 K were considered, as typical values for desert and ocean (SAMUM; Otto et al., 2009). In all cases, a default climatological tropical atmospheric profile (Anderson et al., 1986) and a default $CO_2$ concentration of 420 ppb were considered. For the spectral simulations of the SW
irradiance the Kurucz extraterrestrial solar spectrum (Kurucz, 1994) was used. Simulations were performed in vector mode for DOD at 0.5 μm (hereon referred as DOD) of 0.4, 1 and 1.6, and SZA of 0°, 30°, and 60°. The extinction coefficient profiles that we used in the calculations were derived from the European Space Agency (ESA) - LIdar climatology of Vertical Aerosol Structure for space-based lidar simulation studies activity (LIVAS) database (Amiridis et al., 2015). More specifically, we have used the LIVAS pure-dust extinction coefficient product (Amiridis et al., 2013; Marinou et al., 2017; Proestakis et al.,
2018), as established based on the Cloud-Aerosol Lidar with Orthogonal Polarization (CALIOP) backscatter coefficient and particulate depolarization ratio profiles at 532 nm. The LIVAS pure-dust extinction coefficient profiles in this study are extracted following climatological aggregation of all LIVAS profiles for the months of June, July and August, for 2006 to 2018, over the areas of interest. We should note though, that the extinction coefficient profiles used provide only the vertical distribution of the extinction coefficient, whereas the actual values of the extinction coefficient at each height are scaled
according to the fixed total AOD we use, as mentioned above.

Simulations were performed for all (four) possible combinations of the following cases:

- assuming that particles are spheres or spheroids
- considering that size distribution spans up to 10 μm (i.e., SD10) or up to 50 μm (i.e., SD50)





We also considered two reference cases, one for desert and one for ocean, for aerosol-free skies, providing the "reference
irradiances" above desert and ocean.

The dust radiative effects, under the assumption of different particles shapes (spheres vs spheroids) and sizes (SD10 vs SD50),
have been assessed and they have been expressed as differences in corresponding irradiances, as well as differences in the
integrated values for the corresponding SW and LW spectra. The results are discussed separately for the SW and LW
irradiances, due to the significant differences in the physical mechanisms that control the radiative transfer of light in these
spectral regions. Changes in aerosols practically affect LW irradiance only in the range 8 - 13 μm (atmospheric window). At
shorter and longer wavelengths of the thermal spectrum the absorption from water vapour and $CO_2$ are so strong, that changes
in any other atmospheric constituent have a minor contribution to the overall absorption and emission of the thermal irradiance.
Thus, differences in LW irradiance that were estimated for wavelengths outside the atmospheric window were in all cases
within the uncertainty of the simulations. In the following, differences in SW irradiance refer to the spectral range of 0.35-2.5
μm, differences in the LW irradiance refer to the atmospheric window at 8 - 13 μm, while the total LW irradiance, when
calculated, is the integral of the irradiance in the range of 2.5 - 40 μm.

## 3. Results

### 3.1 Differences in irradiance due to shape

As a first step, the differences in SW and LW irradiances ($F_{spheroids}$ and $F_{spheres}$) due to the different shape of the particles
were calculated. For the calculations we considered only large dust particles (i.e., SD50).

The presented relative differences are calculated as shown in Eq. 2 and Eq. 3.

$$\Delta F = F_{spheroids} - F_{spheres} \qquad\qquad\qquad \text{Eq.2}$$

$$change \; \% = 100\% \cdot \frac{\Delta F}{F_{spheres}} \qquad\qquad\qquad \text{Eq.3}$$

For the number of photons used to perform the simulations (i.e., $10^8$ photons), the estimated statistical noise in the integrals of
the simulated SW is very small (<0.1%) while the noise in the simulated LW is larger (of the order of 1%).





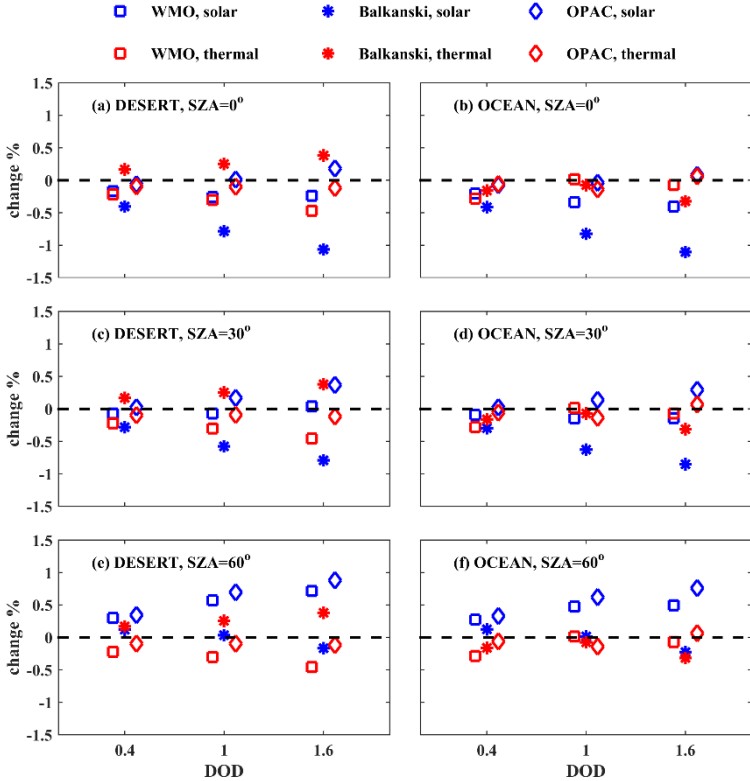

**Figure 8**: Relative differences (in %) for the irradiances in the SW (blue) and LW (red) at BOA, for SD50, considering spherical and spheroidal shapes (see Eq.2). The differences are presented for calculations above desert and ocean, for different RIs of the dust particles, DODs and SZAs.





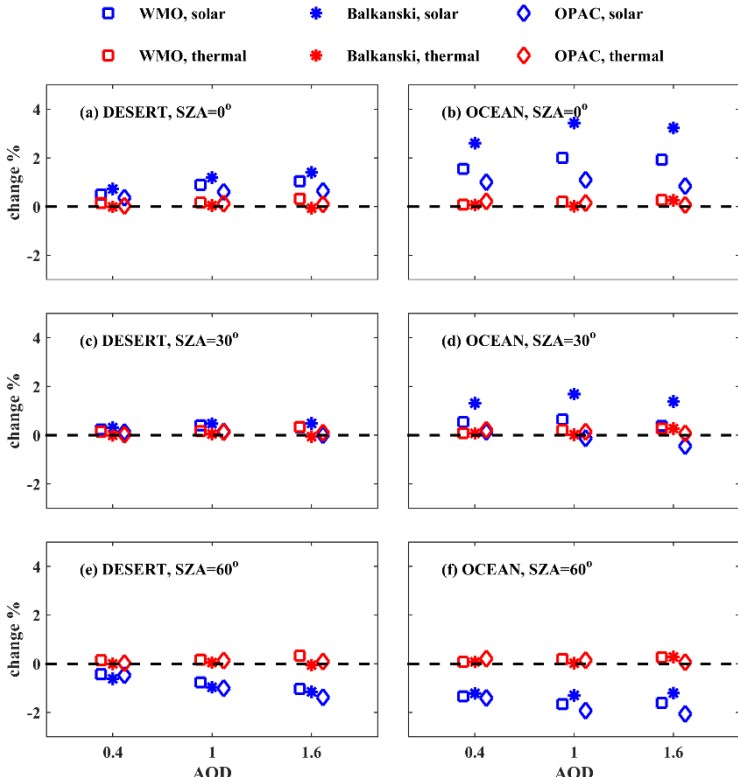

**Figure 9:** Same as Fig. 8, for the relative differences (in %) for the irradiances in the SW (blue) and LW (red) at TOA.

As shown in Fig. 8, for different dust particle shapes, differences below 1.5 % for SW and below 0.5% for LW were found at

BOA, for all conditions considered in the study (i.e., different RI, surface, DOD, and SZA), whereas at TOA the differences were below 4% (Fig. 9). Although differences are small it is still interesting that the sign of differences in the SW can alter depending on the SZA, as well as that differences in the SW increase with increasing DOD. While at BOA differences are similar over the desert and over the ocean, at TOA the differences in the SW are larger over the ocean than over the desert. In the LW the differences are generally within the level of noise. In the SW, using the RI of Balkanski et al. (2007) leads to more

positive differences relative to using the RIs of OPAC and WMO (1983), due to the higher scattering imposed by the former (Fig. 7a).

## 3.2 Differences in irradiance due to size

This section presents the differences in SW and LW irradiances for different SDs considered for dust particles (i.e., SD10 and SD50). We found larger differences relative to the ones related to shape (Sect. 3.1). Spectral differences are also discussed in

this section in order to understand how differences in spectral optical properties affect the results. Equation 4 provides the





absolute difference and Eq. 5 the corresponding relative difference. $F_{SD10}$ is the irradiance for SD10 while $F_{SD50}$ is the irradiance for SD50.

$$\Delta F = F_{SD50} - F_{SD10} \qquad\qquad \text{Eq.4}$$

$$change\ \% = \frac{\Delta F}{F_{SD10}} \cdot 100\% \qquad\qquad \text{Eq.5}$$


### 3.2.1. Spectral differences

In this Section, the absolute and relative differences in the SW and LW irradiances, when SD10 and when SD50 dust particles are considered, are discussed with respect to wavelength. The differences are presented, when dust is above desert, for SZA=30° and DOD of 1.6 (Fig. 10). Changing any of these parameters (i.e., type of surface, SZA, DOD) has a very small

impact on the spectral patterns of the differences and affects mainly their magnitude. It should be kept in mind that in the following figures the x-axis is in logarithmic scale. In spectral regions where the effect of components such as water vapour dominates over the effect of aerosols, the absolute differences are practically zero and the calculated % differences are highly uncertain. Thus, for clarity the differences in the LW are presented only for $8 - 13\mu m$. Differences in upwelling irradiances at TOA and downwelling irradiances at BOA are presented.




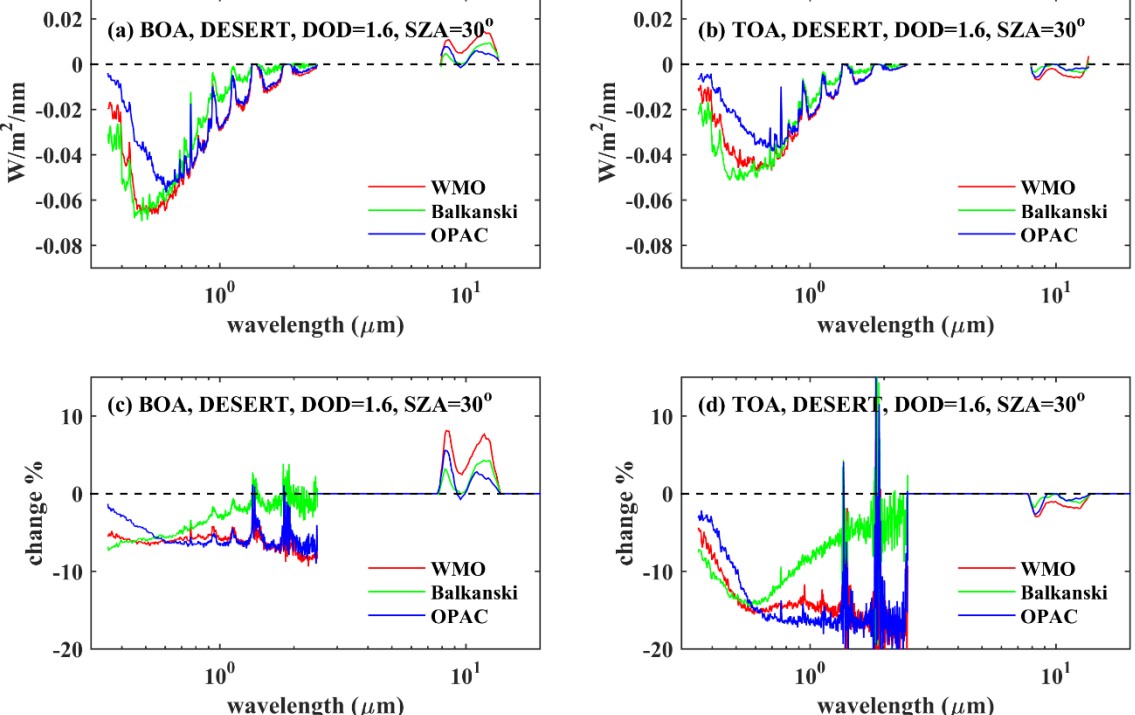

**Figure 10**: Absolute (a,b) and relative (c,d) spectral differences for downwelling irradiances at BOA (a,c) and upwelling irradiances at TOA (b,d), considering SD10 and SD50 dust particles, with spherical shapes. The differences are presented for calculations above desert, for SZA=30° and DOD of 1.6, for different RIs of the dust particles.


As shown in Fig. 6, considering SD50 instead of SD10 results in lower SSA in the SW (i.e., more absorption). The extinction coefficient (and consequently the DOD) varies slightly with respect to wavelength in the SW (Fig. 6a), and thus the relative differences in the SW irradiances at TOA and BOA are strongly anti-correlated with differences in the SSA (Fig. 6g). Increased absorption in the atmosphere due to the presence of larger aerosols results in less SW radiation at TOA and BOA. The absolute

differences in the SW (at TOA and BOA) are determined by the differences in the SSA (anticorrelated) and are proportional to the absolute values of the solar irradiance. In the LW, differences in the irradiances at TOA and BOA are primarily affected by differences in the DOD (Fig. 6a). The DOD spectral dependence is different for SD10 and SD50, and in the range of 8 -13 μm it is larger for SD50 for a given DOD at 0.5 μm. Downwelling LW irradiance increases at BOA and upwelling LW irradiance decreases at TOA. The role of the changing SSA seems to be less important relative to the role of differences in

AOD in the range 8 – 13 μm. Using SD50 for the simulations results in higher DOD in the LW relative to using SD10. Thus, more LW photons are either absorbed or scattered, resulting to decreased upwelling SW radiation at the TOA. Some of the scattered photons (that would be heading towards the TOA for SD10) return to the surface increasing the downwelling radiation at the BOA.





In UV-VIS, at BOA and TOA, the smallest absolute differences were found for the RI used in OPAC. Absolute differences in UV-VIS are similar for the RIs of Balkanski et al. (2007) and WMO (1983), but larger than the ones calculated by the use of OPAC. For SW wavelengths that are longer than ~ 1μm, Balkanski et al., (2007) consider that absorption is insignificant (either for SD50 or for SD10, (see Fig. 6) which results in the smallest absolute and relative differences in irradiance for SD10 and SD50. For the LW the absolute and relative differences (positive for BOA, negative for TOA) are generally larger when the RI of WMO (1983) is used.

**3.2.2. Differences in SW and LW irradiances**

The relative differences in the SW and LW irradiances (calculated from the integrated irradiances in 0.35 – 2.5 μm for SW and 2.5 – 40 μm for LW) at BOA are shown in Fig. 11, while the corresponding absolute differences are shown in the Appendix (Fig. A1). The same differences for TOA are shown in Fig. 12 (and in Fig. A2 in Appendix), respectively. The differences are presented for desert and ocean with respect to SZA and DOD. As shown in the figures, they are generally larger for SW relative

to LW irradiances (with the differences in the LW to be comparable to the level of noise), especially over the desert.

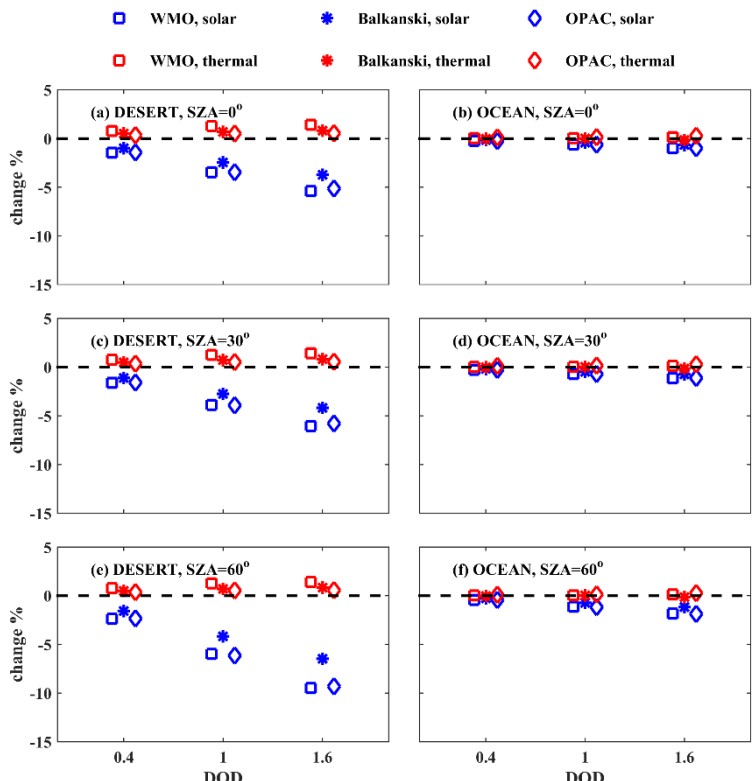



**Figure 11:** Relative differences (%) between the irradiances in the SW (blue) and LW (red) at BOA, considering SD10 and SD50 dust particles, with spheroidal shapes (Eq. 5). The differences are presented for calculations above desert and ocean, for different RIs of the dust particles, DODs and SZAs.


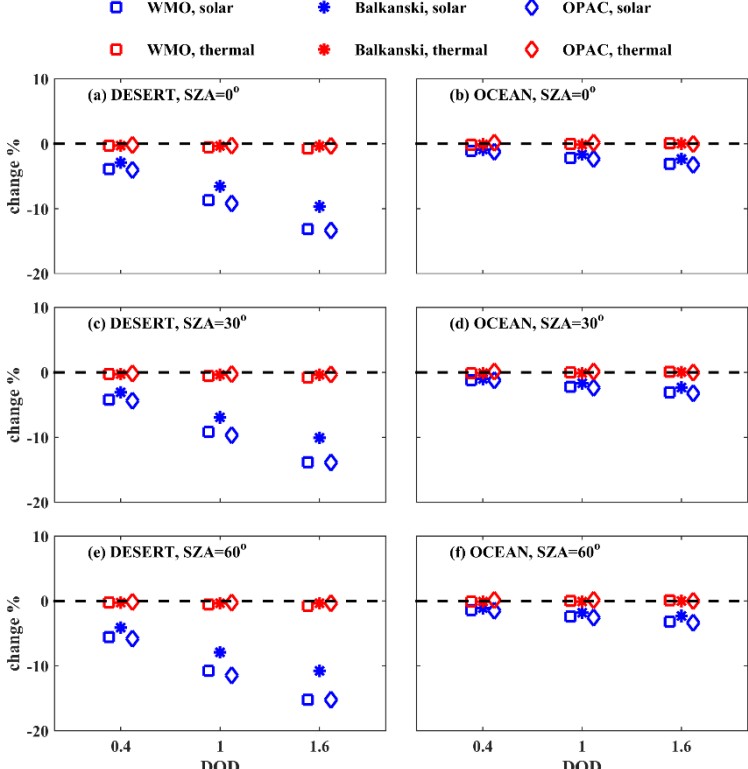

**Figure 12:** Similar to Fig. 11 for TOA.

Over the ocean, considering large dust particles results in a reduction in the SW irradiance, which however is less than 2% (or 5 W/m$^2$) at BOA and less than 3% (or 5 W/m$^2$) at TOA. The differences over the ocean are smaller relative to the desert, mainly due to the fewer dust particles with diameters larger than 10 μm above the ocean (see Fig. 2) which leads to very small differences in aerosol optical properties (see Fig. 5, 6 and discussion in Sect. 2.2). An additional reason for the weaker effect of large dust particles over the ocean (though less significant in this case), is the lower surface albedo of the ocean relative to the desert, since the impact of absorbing aerosols becomes stronger above brighter surfaces.

Over desert, the reduction in SW irradiance at BOA can be up to ~ 5% (or ~50 W/m$^2$) for DOD of 1.6 and SZA=0°, and up to ~ 10% (~30 W/m$^2$) for DOD of 1.6 and SZA=60°. As SZA increases, absolute differences in SW at BOA decrease (Fig. S1) while relative differences increase (Fig. 11). Larger SZAs correspond to longer paths of the SW photons in the layer of dust,





and subsequently to increased attenuation. Thus, the relative contribution of large dust particles in attenuating the SW irradiance also increases, leading to larger relative differences. Nevertheless, at larger SZAs, less downwelling solar radiation

reaches the aerosol layer because it has been already attenuated in the atmosphere. Subsequently, less photons interact with aerosols, resulting in smaller absolute differences. At TOA, the reduction in SW irradiance is larger relative to BOA and the role of SZA becomes less significant. The reduction in the outgoing SW irradiance at TOA for DOD of 1.6 is 15 - 18% (~20 - 30 W/m$^2$).

The smallest differences (up to ~3% and ~5% at BOA and TOA, respectively) among the three RIs were calculated for the RIs

reported in Balkanski et al. (2007), for which the difference in the SSA between SD10 and SD50 particles is the smallest for SW wavelengths longer than 1μm (Fig. 6).

Regarding the calculations in the LW, aerosols scatter and absorb part of the radiation emitted by the Earth's surface, that would exit the atmosphere, with part of the scattered and (absorbed and then) emitted radiation returning to the surface. When large dust particles are considered, the scattering and absorption of the radiation is larger, mainly over the desert (Fig. 5 and

6). Thus, the LW radiation that returns to the BOA increases (Fig. 11) and the LW radiation that exits the atmosphere (i.e., at TOA) decreases (Fig. 12). At BOA the increase is however below 2% for desert and below 0.5% for ocean. At TOA the outgoing LW irradiance decreases by less than 1% over the desert (and even less over the ocean) when large dust particles are considered, which, as has been already discussed, is within the noise of the simulations.

### 3.3. Direct radiative effects

In this section, the effects of shape and size on the DRE of dust over the desert and the ocean are discussed for different SZAs. For each of the three RIs, the SW and LW DREs at TOA and BOA were calculated. First, the net irradiances were calculated for BOA and TOA ($F_{NET,BOA}$ and $F_{NET,TOA}$ respectively), using Eq. 6:

$$F_{NET} = F^{\downarrow} - F^{\uparrow} \qquad\qquad \text{Eq.6}$$

Where $F^{\downarrow}$ and $F^{\uparrow}$ are the downwelling and upwelling irradiances, respectively.

Then, the direct radiative effect (DRE) was calculated as the difference between the test ($F_{NET,TEST}$) and the reference ($F_{NET,REF}$) net irradiances, as shown in Eq. 7:

$$DRE = F_{NET,TEST} - F_{NET,REF} \qquad\qquad \text{Eq.7}$$

In this study, the reference net irradiance corresponds to aerosol-free atmosphere while the test net irradiance corresponds to the same conditions when aerosols are considered.





### 3.3.1. Direct Radiative Effects for different model inputs

In this section we present the DRE of dust particles when considering SD50 and spheroidal shapes (more realistic representation of dust), and SD10 and spherical shapes (less realistic representation of dust, used in current dust models).

Figures 13 and 14 present the calculations for DOD of 1 for different RIs, for BOA and TOA, respectively. The direct radiative effects were calculated for the SW and LW, as well as for the TOTAL (SW+LW) irradiances.

#### DRE at BOA

In the following discussion we must keep in mind that differences shown Fig. 13 (DRE at BOA) are positive when more
radiation reaches the Earth's surface, while differences in Fig. 14 (DRE at TOA) are positive when less radiation exits the atmosphere.

At BOA the TOTAL DRE ranges between ~-200 and -100 W/m$^2$ for different dust properties and SZAs between 0° and 60°. Using more realistic aerosol properties leads to less TOTAL radiation at BOA. The results are similar over the desert and the ocean. For more realistic aerosol properties the DRE is less negative over the desert (by up to 25% for the RI of Balkanski et
al. (2007) at SZA=0°) and over the ocean, although the difference is small, less than 5% over the ocean. Differences are mainly due to the used RIs and SZA rather than differences in size and shape.





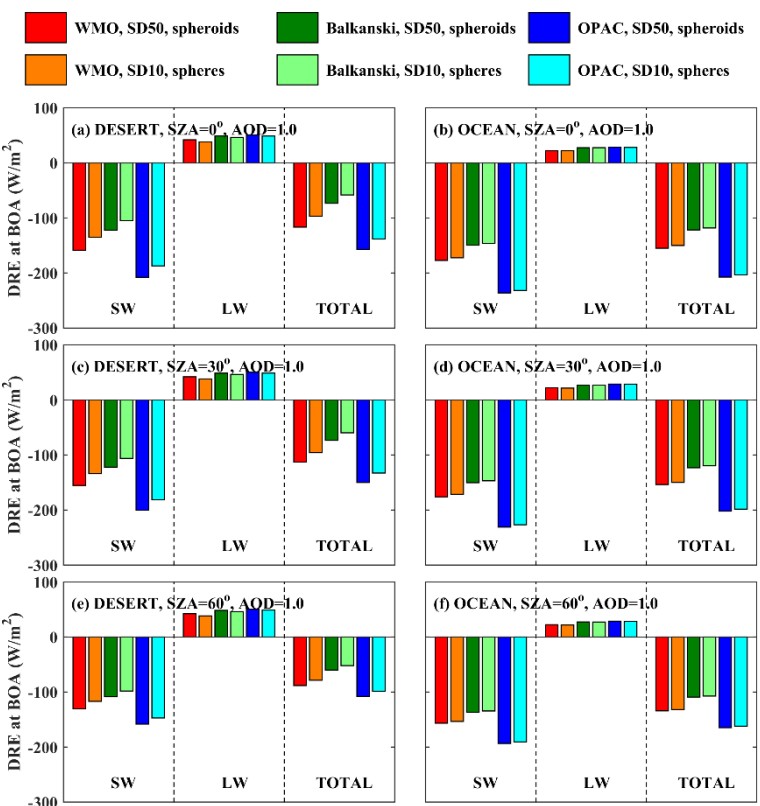

**Figure 13:** DRE at BOA for different RIs and SZA, for SD10 with spherical shapes ($DRE_{SD10, \text{ spheres}}$) and SD50 with spheroidal shapes ($DRE_{SD50, \text{ spheroids}}$).

### DRE at TOA

Over the desert, the presence of dust particles results in all cases in heating of the Earth-atmosphere system considering the TOTAL radiation at TOA, the magnitude of which however depends strongly mainly on the used RI and SZA, but also on SD (as discussed in Sect. 3.3.2). While the DRE differences for different RIs are small for the LW and independent from SZA (as expected, since the LW radiation is mainly emitted from the Earth's surface), the SW DRE depends strongly on the used RIs and changes significantly with SZA (Fig. 14). Negative SW RF of $\sim$-20 W/m$^2$ was calculated for the RI provided by Balkanski et al. (2007), at SZA=60°, when SD10 spheres were considered. Differences of 15 – 25 W/m$^2$ were found for the TOTAL DRE that was calculated for more (SD50 spheroids) and less (SD10 spheres) realistic dust optical properties. Using different RIs results in very large differences, up to 80 W/m$^2$ in TOTAL DRE.

Over the ocean, the sign of the DRE also depends on SZA and the used RI, whereas the overall effect of SD and shape is minor due to the small contribution of larger dust particles in SD50. The DRE in the SW is always negative (i.e., increase of the outgoing SW radiation at TOA resulting in cooling of the Earth-atmosphere system), with increasing absolute value as the





SZA increases. The DRE in the LW is positive (i.e., warming) and independent from SZA. As a result, the TOTAL DRE is positive and negative for SZA=0° and 30°, for different RIs, and it is negative for SZA=60°. Differences in RF up to ~40 W/m$^2$ due to different RIs were calculated over the ocean.

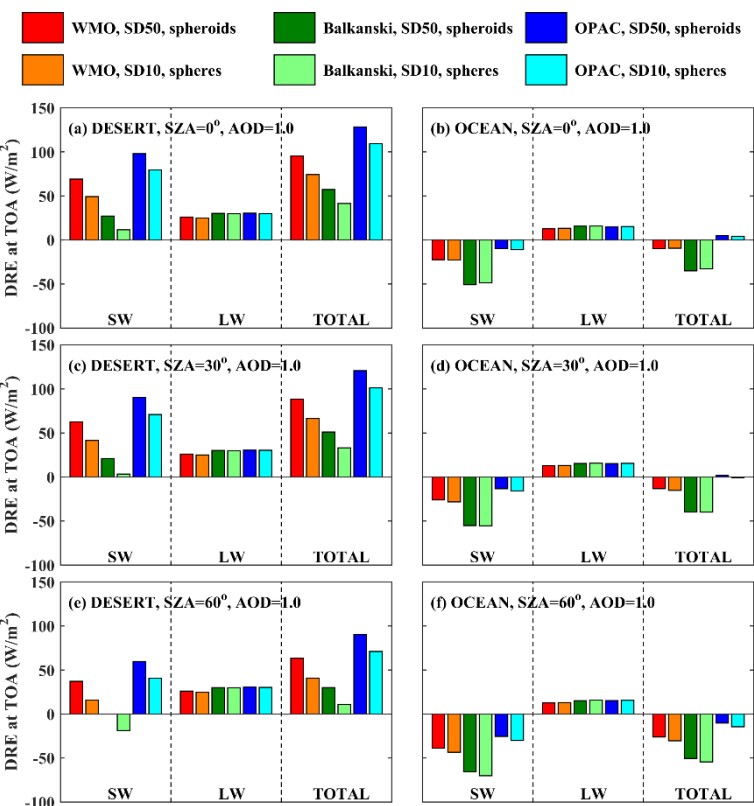

**Figure 14:** Same as Fig. 13 for DRE at TOA.

By comparing the results in Fig. 13 and 14 we see that the impact of the used RI for the calculation of DRE at BOA is less significant than for TOA (at least in terms of fractional differences). Nevertheless, the results presented in Fig. 13 and 14 show that the used RI is still the main uncertainty factor for the calculation of the DRE.


**3.3.2. Effects of size and shape**

The present section aims to show and compare the effects of using more realistic size and shape on DRE. As discussed in the previous sections, for given RI, differences in the irradiances at BOA and TOA that are significantly larger than the uncertainty in the simulations emerge mainly when we compare the effect of using different SDs over desert. This is also depicted in the

DRE calculations presented here.





For different RIs, the differences in DRE due to the size of the particles (when considering SD50 versus SD10) above desert change up to 12 W/m² at BOA (decrease) and TOA (increase), as shown in Fig. 15 and 16, respectively. The differences are much smaller above the ocean. At BOA (Fig. 15), considering SD50 instead of SD10 results in less SW and TOTAL radiation (more cooling at the surface), above the desert and the ocean. For the LW radiation, the DRE at BOA increases when
considering larger particles (SD50). This does not happen when considering the RI provided in Balkanski et al. (2007) (for all SZAs) and in WMO (1983) (for SZA=0°) over the ocean, where very small decreases were found (<1 W/m²). The decrease of the TOTAL radiation at BOA (i.e., more cooling at the surface) due to larger particles, generally decreases with SZA. For the studied cases, the corresponding differences in Fig. 15 range between 3 and 11 W/m² over the desert, and between 1 and 2 W/m² over the ocean.

At BOA, using spheroids instead of spheres results in smaller DRE due to TOTAL radiation at SZAs 0° and 30°, and greater DRE due to TOTAL radiation at 60°, with the differences being much smaller relative to differences due to size (<5 W/m²). However, there are specific cases where the effect of shape is comparable to the effect of size. For example, for SZA=0° over the desert, when considering the RI provided by Balkanski et al. (2007), the difference in TOTAL DRE due to size is ~ 6 W/m², while the difference due to shape is ~ 2 W/m². Over the ocean, for the same case, the differences are ~ 2 W/m² and ~ 6
W/m², respectively. This case indicates that, although at TOA the uncertainties in climate model simulations due to size are clearly dominant over the uncertainties due to shape, at BOA the effect of shape may be larger under specific conditions, although the differences will still be small, less than 5 W/m² (for AOD=0.4).

Different SZAs do not practically affect the differences due to size at TOA, but they have a stronger impact at BOA. Differences due to shape change however with changing SZA at both, TOA and BOA. For example, as the SZA increases
from 0° to 60° the differences due to shape change from negative to positive at TOA (Fig. 16), although in all cases they are small.




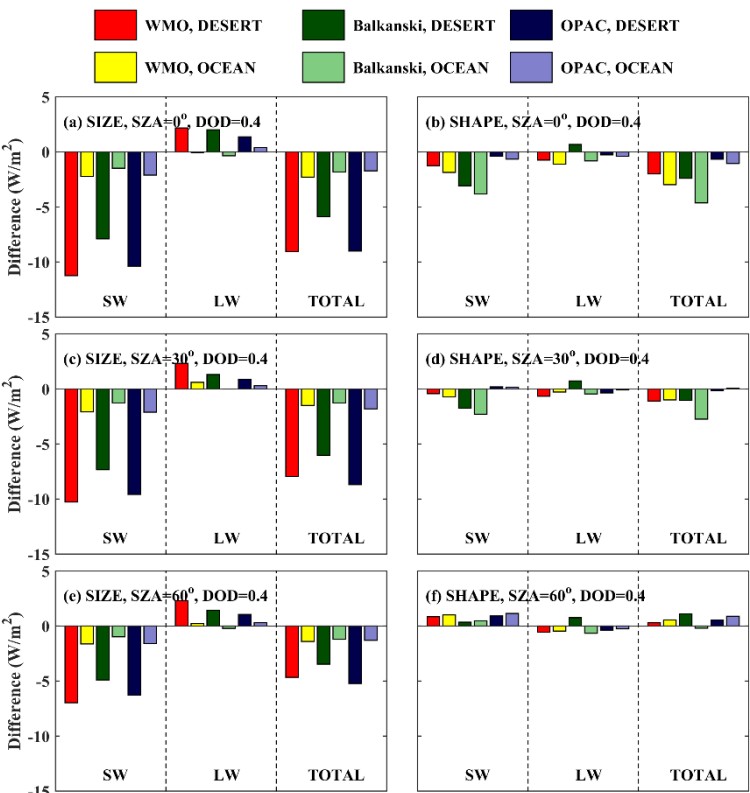

**Figure 15:** Absolute differences in DRE at BOA for DOD of 0.4, for different shape and size of the dust particles. Left column: absolute
differences in DRE due to size ($DRE_{SD50} − DRE_{SD10}$). Right column: absolute differences due to shape ($DRE_{SD50, spheroids} − DRE_{SD50, spheres}$).

At TOA, considering SD50 instead of SD10 results in less SW, LW, and TOTAL radiation leaving TOA, thus in more warming
of the Earth-atmosphere system, with the differences being larger over the desert relative to the ocean by about an order of
magnitude. The corresponding plots with the relative changes instead of the absolute changes are shown in the appendix (Fig.
A3 and A4 for BOA and TOA, respectively). Relative changes due to size at TOA can be as high as 300%, confirming the
findings of previous studies (e.g., Otto et al., 2011), which however correspond to small differences, of less than 3 W/m$^2$, in
absolute numbers.

From Fig. 16 it is clear that at TOA above desert the effect of size on DRE is dominant over the effect of shape. Above the
ocean, the effects of size and shape are comparable, but in all cases the differences are below 3 W/m$^2$. As already discussed
in Sect. 3.3.1, it is also clear from Fig. 15 and 16, that the estimated impact of using less accurate size or shape for the
simulations depends strongly on the used RI, i.e., on the composition of the dust particles.





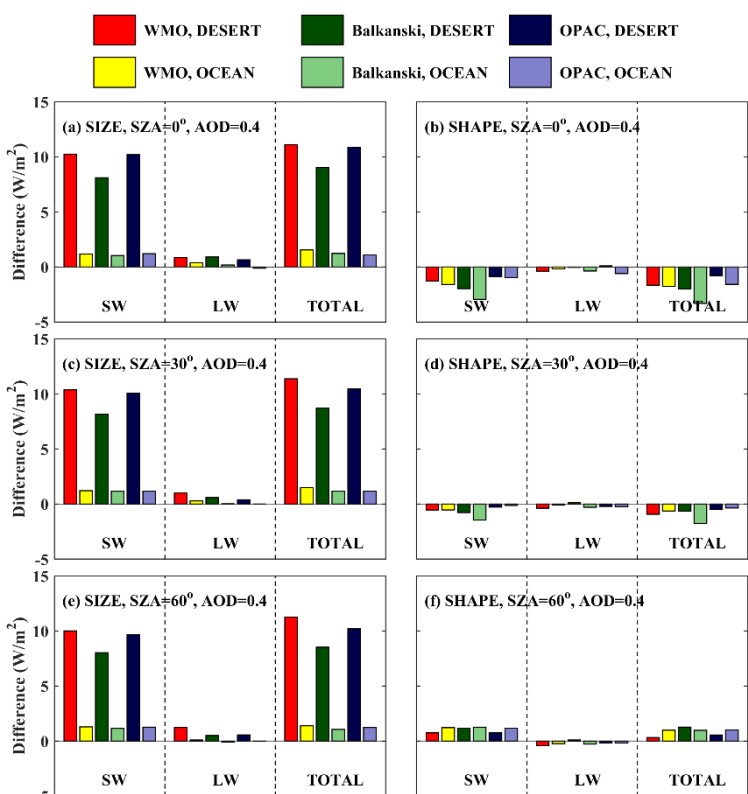

**Figure 16:** Same as Fig. 15 but for TOA.

In this section, results are presented for DOD of 0.4 because it is more representative for typical (and not extreme) dust events, and the results are directly comparable with the results from other studies (e.g., Otto et al., 2011). The corresponding differences for DOD of 1 and 1.6 are presented in Fig. S5 and S6 for BOA and Fig. S7 and S8 for TOA in the appendix. Increasing the DOD from 0.4 to 1 results in differences that are nearly double compared to those presented in this section, while increasing DOD from 0.4 to 1.6 results in differences that are nearly triple.

**4. Comparison with other studies**

The radiative effect of very large dust particles is usually underestimated in dust models. In fact, most dust models completely omit these particles. The underestimation is high closer to the desert dust sources ($\sim 10$ W/m$^2$ for the SW irradiances at BOA (cooling is underestimated) and TOA (warming is underestimated) for DOD of 0.4). It is lower further away from the dust sources, above the ocean, where even for DOD=1.6, the highest calculated underestimation is below 5 W/m$^2$. This is due to

the lower fraction of large dust particles (with diameters $10 - 50$ μm) over the ocean, with the fraction to be more than two



orders of magnitude lower than above desert (Ryder et al., 2019). We should note though that the fraction of the large dust particles in the dust mixture over the ocean has been reported to be larger by other studies in the literature (e.g., Otto et al., 2009; 2011) than the one reported in Ryder et al. (2019) and used here.

The additional warming of the Earth-atmosphere system due to the large dust particles above desert, is generally in agreement
with Ito et al., (2021) who showed that over major emission sourse regions (e.g., the Sahara Desert) using more realistic optical properties results in less net downwelling radiation reaching the BOA and less upwelling radiation at TOA, i.e., more atmospheric warming. The findings of our study also agree with the results of Kok et al. (2017) who showed that more realistic representation of aerosols in global climate models would lead to less radiation leaving TOA. The findings of the latter study are not however directly comparable with our findings since they refer to global scale.

The effect of dust size on aerosol optical properties has been discussed in Ryder et al. (2019) who (as in this study) used data from the Fennec campaign, as well as from the AER-D campaign and the RIs suggested by Colarco et al. (2014) and Balkanski et al. (2007). The fraction of the large dust particles in the dust mixture over the ocean reported in Ryder et al. (2019) was also significantly lower relative to the ones reported for other campaigns (e.g., Otto et al., 2009; 2011). They found that excluding giant particles over the Sahara Desert results in underestimation of both SW and LW extinction, by 18 % and 26 % respectively.
They also estimated smaller but non-negligible differences over ocean. Their findings are consistent with the magnitude of the estimated differences for the three RIs considered in this study. We moved on, a step further relative to Ryder et al. (2019), and quantified the effect of using more realistic SD and particle shapes for dust on the simulated irradiances and DREs at the TOA and the BOA.

We also confirm the findings of Song et al. (2022) who report that the used RI is the main uncertainty factor in the modelling
of dust DRE. In the latter study the uncertainty due to RI is estimated to be ~45%. We show that under certain conditions, differences in the calculated DRE for different RIs can be much larger, exceeding 100%. Song et al. (2022) also report that on a global scale the second most significant factor of uncertainty is the used SD while the role of particle shape plays a negligible role. Our results verify the latter as well for the studied cases.

Otto et al. (2009) used data from the SAMUM campaign and showed that dust leads to cooling over the ocean but warming
over the desert at TOA due to differences in their spectral surface albedo and surface temperature. In this study we show that the magnitude of the corresponding cooling and warming depends strongly on the used RIs, which again is consistent with the findings of Song et al. (2022). We further show that the use of specific RIs under specific conditions can result even in slight cooling at TOA over the ocean.

Otto et al. (2011) used spheroids instead of spheres for the modelling of the radiative effects of dust particles and estimated
excessive cooling of ~55% over ocean, a very weak change of ~5% over desert, and cooling of ~15% at BOA over both, desert and ocean (considering SZA=0° and DOD of ~0.4 at 0.5 μm, based on data from the SAMUM campaign). We also found a very weak change in RF at TOA over the desert when spheroids were used instead of spheres. Over the ocean, we calculated cooling of the atmosphere, which in terms of relative difference was large in some cases, but it always corresponded to very small absolute differences of 5 W/m² or less.



**5. Summary and conclusions**

Our study focuses on quantifying the radiative effect of the underestimation of size of dust particles and the misrepresentation of their RI and shape in models. We show that although the effect of the dust size is substantial for the radiative impact of dust, the main uncertainty factor is the RI (i.e., the chemical composition) of dust, which affects both, the magnitude of the DRE and the differences in DRE due to size. Using a more realistic shape (spheroids instead of spheres) for the dust particles generally plays a minor role relative to all other factors, but under specific conditions the differences may be comparable to the differences due to the size of the particles (although they are always small, <5 W/m$^2$).

Regarding the spectral shape of differences in irradiance at BOA and TOA due to the use of more realistic shapes and SDs for the dust particles, again the differences strongly depend on the used RI. For given DOD, the differences in the SW irradiances at TOA (upwelling) and BOA (downwelling) depend strongly on the change of the SSA. The differences in the LW are practically significant only in the region 8 – 13 μm and depend mostly on the changes in extinction coefficient (i.e., the DOD) with wavelength.

We showed that the SZA also plays a significant role regarding the radiative effects of dust, and different SZAs may even result in different sign in DRE. At BOA, at SZA=0°, as in Otto et al. (2011), we calculated similar differences over the desert and the ocean when spheroids are considered instead of spheres, i.e., excessive cooling at TOA up to ~15%. However, as the SZA increases, the decrease in RF weakens, and at SZA=60° using spheroids instead of spheres results in positive changes in DRE at BOA. Nevertheless, it should be kept in mind that absolute differences were small in these cases.

From our analysis it is also clear that in the case of dust the LW DRE at TOA should not be considered negligible, as also stated by Sicard et al. (2014). We showed that over highly-reflective surfaces the LW DRE contributes significantly in the warming at TOA and the cooling at BOA. Over darker surfaces (i.e., the ocean) warming due to LW RF counterbalances cooling due to SW RF. LW RF can be even dominant at low SZAs resulting in TOTAL warming above the ocean.

In our study we tried to estimate the effect of using more realistic shapes and SDs for the quantification of the dust DRE close and away from a dust source and investigated the sensitivity of our findings to the SZA and the chemical composition of dust (RIs). It is an effort to show how better parameterization of dust in the models can improve the modelling of the DRE, and which parameters are the most important regarding the uncertainties they introduce. This analysis is not exhaustive, since there are many feedbacks and mechanisms that should be taken into account in order to accurately quantify the DRE. For example, excessive absorption of SW radiation in the atmosphere results in warming, and subsequently in emission of LW radiation by the atmosphere. Excessive LW radiation at the Earth's surface results in warming of the surface (i.e., changing temperature), and subsequently in differences in the spectral shape of the LW radiation emitted by the surface. Such phenomena have not been taken into account in our study.

Further investigation, which is out of the scope of the present study, is necessary to quantify how interactions between solar and thermal radiation, and different aerosol species would alter the radiative impact of dust particles. For example, in our study we neglect the presence of scattering sea-salt particles, accumulated within the marine boundary layer, below the overlying



absorbing dust particles when these are advected over maritime downwind areas. Under these conditions, it is expected an enhancement of the SW atmospheric warming, depending on the dust burden intensity, thus affecting the sign at TOA (weaker cooling or even warming effect). Furthermore, according to Ryder et al. (2021), despite the usually dry conditions during dust events, the presence of dust favours the formation of water vapor in the SAL, which in turn increases the positive LW DRE at TOA, and decreases the LW DRE at BOA.

**Data Availability**

The LIVAS dust products are available upon request from V. Amiridis (vamoir@noa.gr), E. Proestakis (proestakis@noa.gr), and/or Eleni Marinou (elmarinou@noa.gr).

**Author contributions**

Conceptualization: SK, VA, and AT; Methodology: I.F., and A.T.; Formal analysis: IF, AT, MT, and AN; Software: I.F., A.T., M.T., and A. N.; Validation: SK, AG, KP, VB, CE, BM; Investigation: IF, and AT; Resources: AT and VA; Data curation, IF, AT, MT and MP; Visualization: IF, and AT; Writing - Original draft preparation: IF, and AT; Supervision: SK, and VA; Writing - review and editing: all authors. All authors gave final approval for publication.

**Acknowledgements**

Authors would like to acknowledge the Action Harmonia CA21119 supported by COST (European Cooperation in Science and Technology).

**Funding:** This research was supported by D-TECT (Grant Agreement 725698) funded by the European Research Council (ERC) under the European Union's Horizon 2020 research and innovation program. This research was financially supported by the PANGEA4CalVal project (Grant Agreement 101079201) funded by the European Union. E. Proestakis was supported by AXA Research Fund for postdoctoral researchers under the project entitled "Earth Observation for Air-Quality – Dust Fine-Mode - EO4AQ-DustFM".

**Competing interests:** At least one of the (co-)authors is a member of the editorial board of Atmospheric Chemistry and Physics. The peer-review process was guided by an independent editor, and the authors have also no other competing interests to declare.

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



**Appendix**

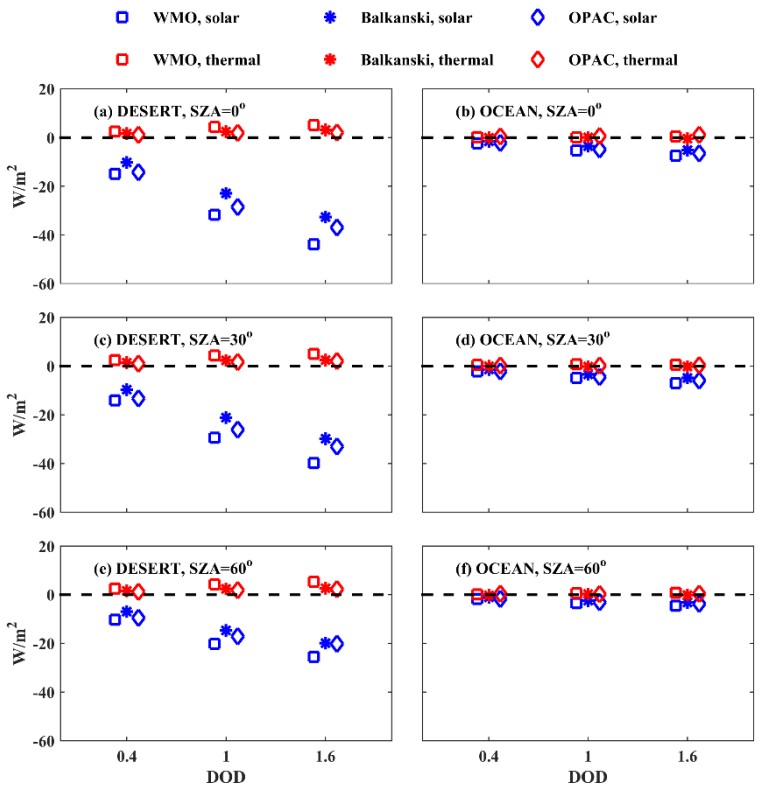

**Figure A1:** Absolute differences between the irradiances in the SW (blue) and LW (red) at BOA, considering SD10 and SD50 dust particles, with spheroidal shapes ($F_{SD50,spheroids}$-$F_{SD10,spheroids}$, Eq. 4). The differences are presented for calculations above desert and ocean, for different RIs of the dust particles, DODs and SZAs.



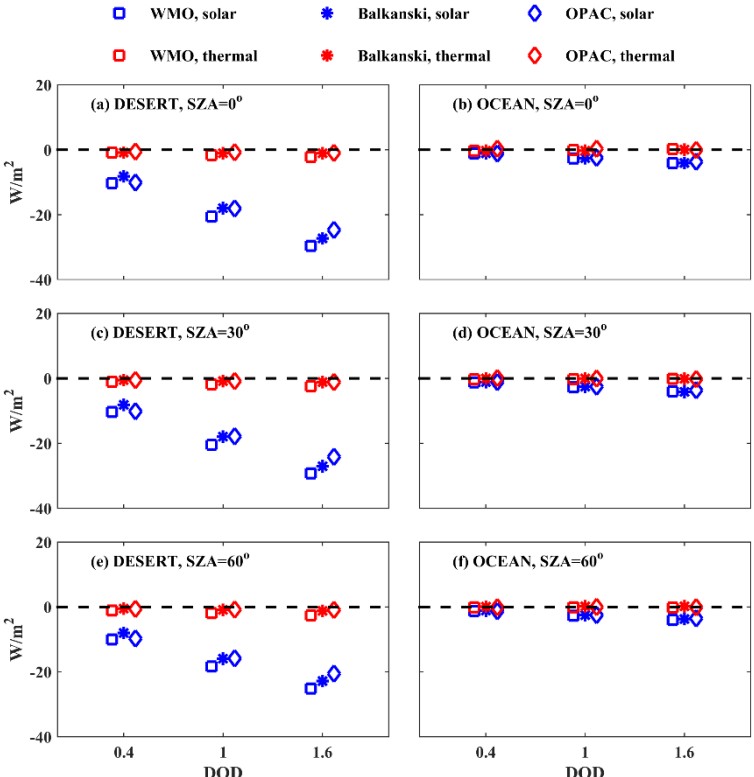

**Figure A2:** Similar to Fig. A1 for TOA.



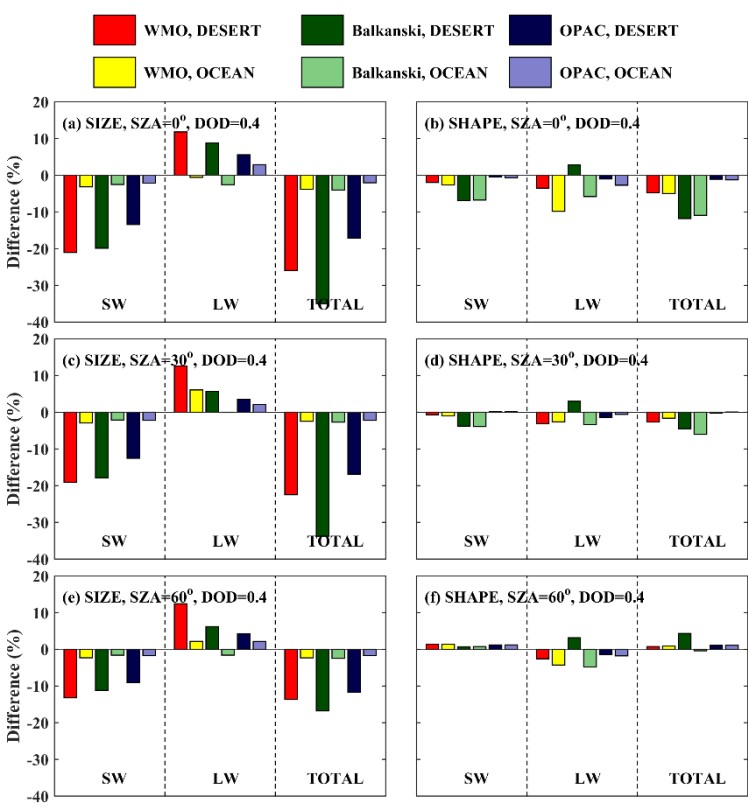

**Figure A3:** Relative differences (%) in DRE at BOA for DOD of 0.4, due to shape and size of the dust particles. Left: relative differences in DRE due to size (DRE$_{SD50}$ − DRE$_{SD10}$). Right: relative differences due to shape (DRE$_{SD50, spheroids}$ − DRE$_{SD50, spheres}$).




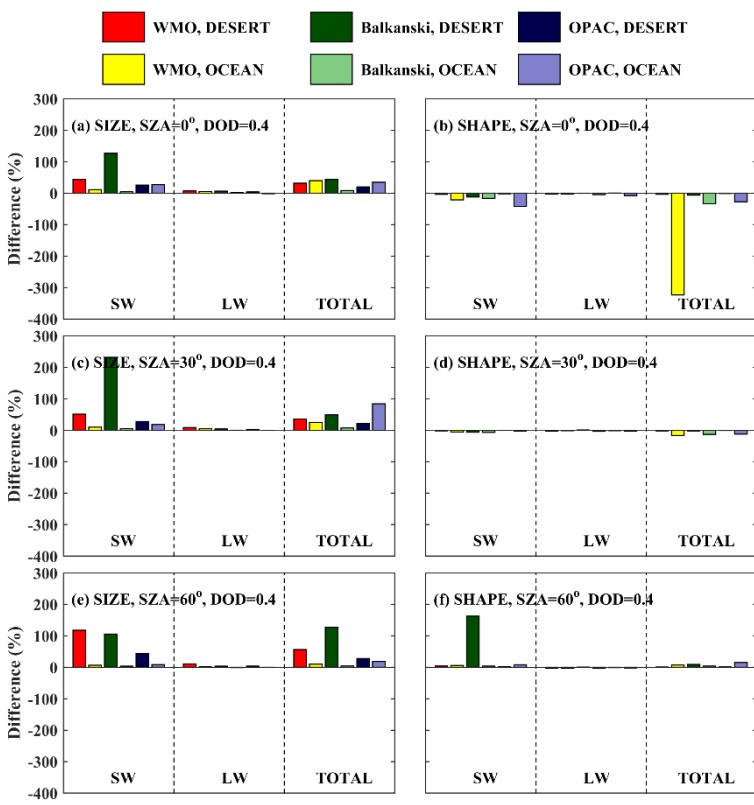


**Figure A4:** Same as Fig. A3 for TOA.





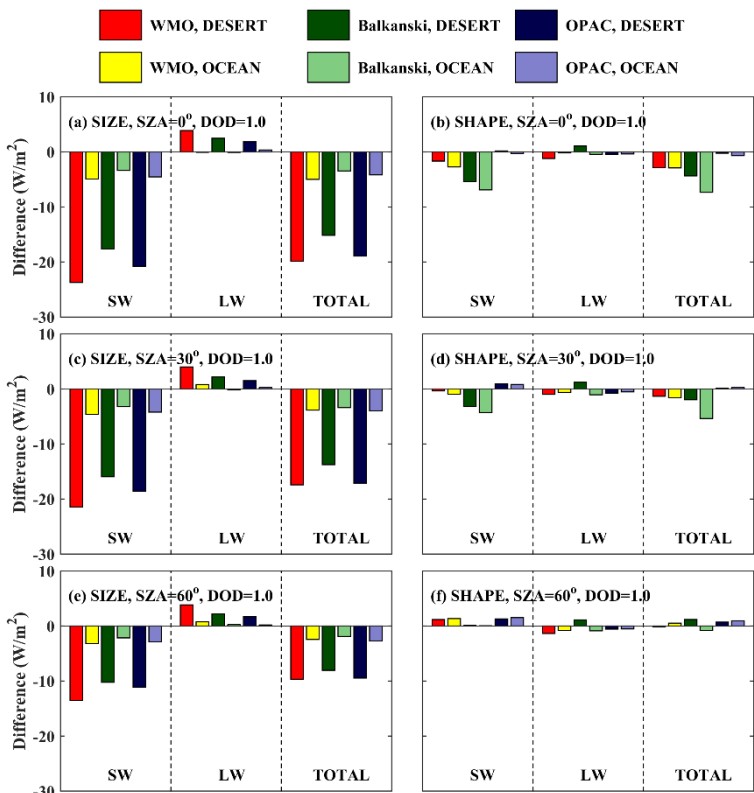

**Figure A5:** Absolute differences in DRE at BOA for DOD of 1, due to shape and size of the dust particles. Left: absolute differences in DRE due to size ($DRE_{SD50} - DRE_{SD10}$). Right: absolute differences due to shape ($DRE_{SD50, \text{spheroids}} - DRE_{SD50, \text{spheres}}$).





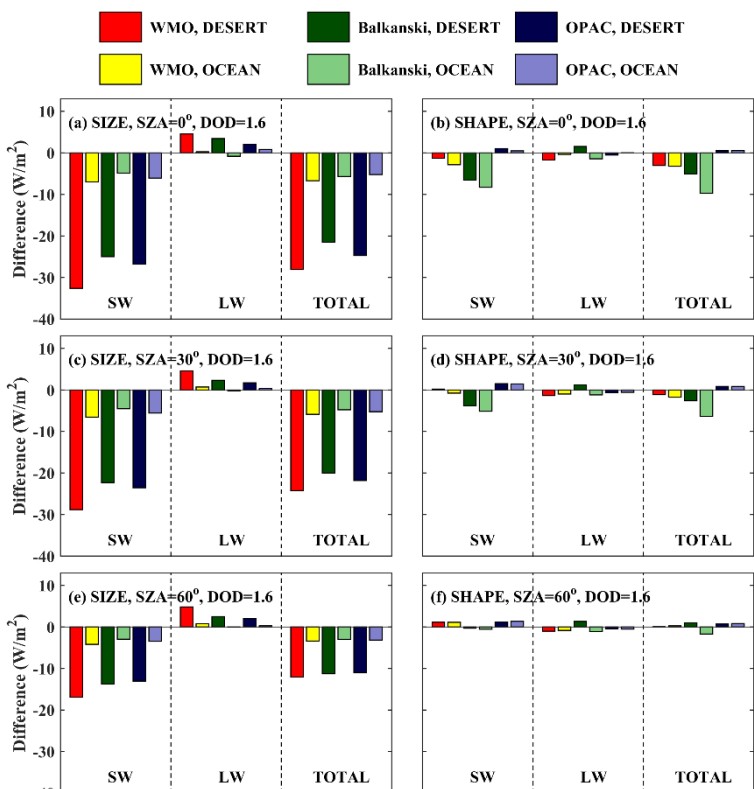

**Figure A6:** Similar to Fig. A5 for DOD of 1.6.




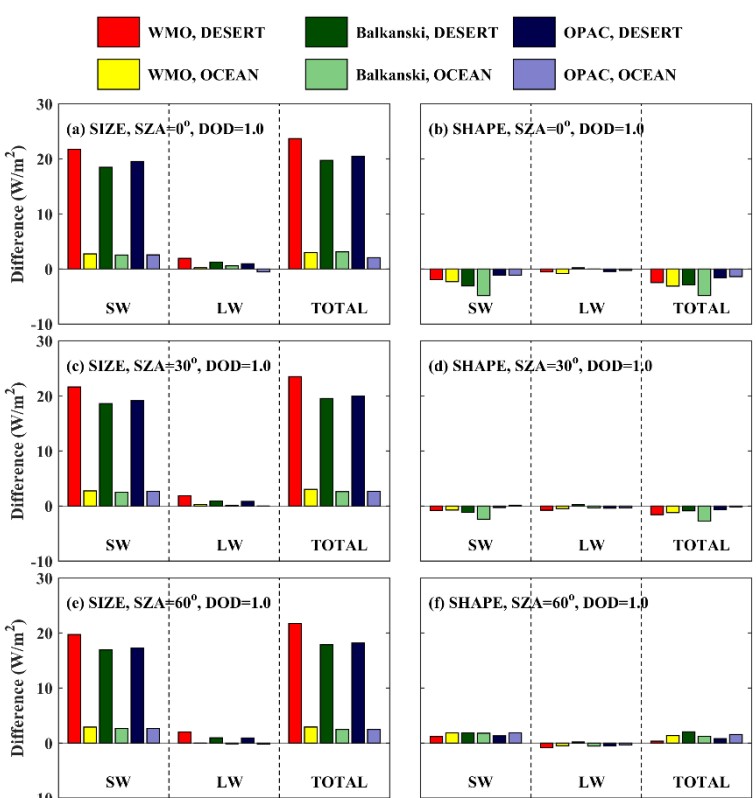

**Figure A7:** Absolute differences in DRE at TOA for DOD of 1, due to shape and size of the dust particles. Left: absolute differences in DRE due to size ($DRE_{SD50} - DRE_{SD10}$). Right: absolute differences due to shape ($DRE_{SD50, \text{spheroids}} - DRE_{SD50, \text{spheres}}$).





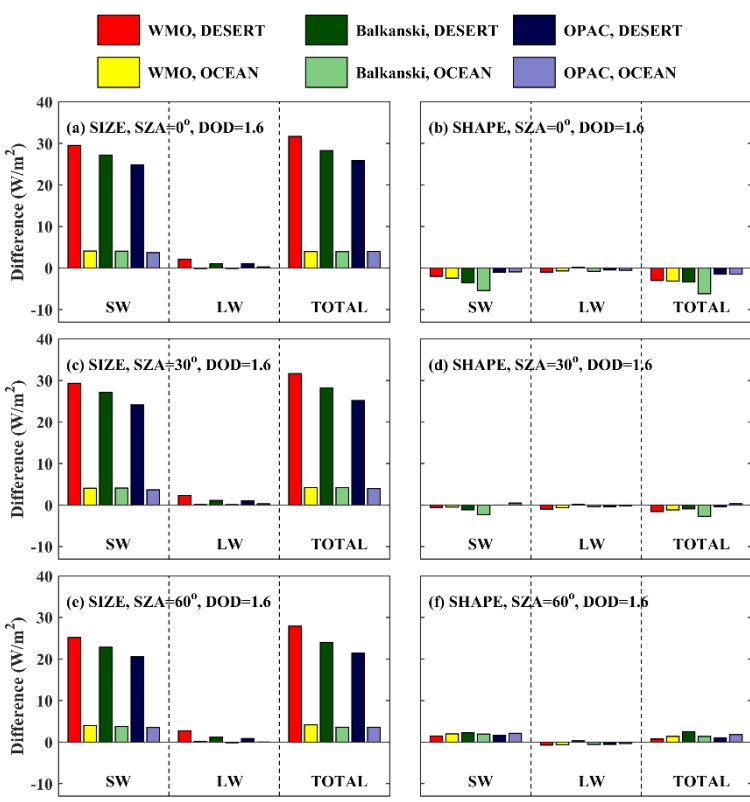

**Figure A8:** Similar to Fig. A7 for DOD of 1.6.
