# Peer review of "A sensitivity study on radiative effects due to the parameterization of dust optical properties in models"

_EGUsphere, 2023_

## Referee Comment (RC3)

Review on "**A sensitivity study on radiative effects due to the parameterization of dust optical properties in models**" by **Fountoulakis et. al.**

This paper presents a sensitivity study on radiative effects due to Saharan dust aerosols. The parameter tested in the sensitivity study are 1) optical properties of aerosols under three different spectral-resolved refractive indexes (WMO 2023, OPAC: Köpke et al. 1998, 2015 and Balkanski et al. 2007). 2) Size range of aerosol size are: 0.1 - 10 µm (as done in most of former and current models) and 0.1 - 50 µm. 3) Shape form (spherical or spheroidal shapes) Calculations are made over the desert and over the ocean. The sensitivity is first led on macroscopic aerosol radiation properties (extinction, scattering and absorption coefficients), then thanks to simulations with the MYSTIC radiative transfer model (Mayer, 2009) on radiative fluxes, and at the end, the sensitivity on radiative forcing is computed and analysed.

The paper has to be understood and evaluated as an advance for radiative transfer simulations in atmospheres containing dust aerosols but not about generalities concerning dust aerosol DRE, in contrary to the wrong understanding of anonymous author of RC2 comment. In this way, the paper answers and quantifies some open questions of radiative transfer in atmosphere containing dust aerosols, a domain that needs such kind of detailed and structured studies, since the difficulty of parametrizing the radiative transfer equation in radiative transfer models for the case of dust aerosol is contained in the lack of order of magnitude and on the unknowledge about the influence of the different parameters (especially RI, size distribution and shape). Therefore, this study is a significant advance in this topic thanks to the clarifications and the quantifications that it brings. Thus, this paper is worth to be published in Atmospheric Chemistry and Physics. The paper is welled structured, the radiative transfer simulations selected are relevant, the method and the results are well explained and well presented. The paper itself is well presented, and very clear, as well from a didactic as from a linguistic aspect. Figures and tables, are presented in a clear and ergonomic way, and the results of the simulations shown in the figures are analysed in the text of the manuscript in a meticulous way.

For all these reasons, I kindly recommend to accept this article in Atmospheric Chemistry and Physics, after the minor corrections I suggest, and after the authors briefly answer the few questions here below.
If the paper will be (as I suggest it) accepted with minor corrections, please give some explanations in answers to my comments

**Major comments/questions**

1) Please add an acronym table to define and summarize in the same place the main used acronyms (RI, TOA, BOA, AOD, DRE, IRE, SRE, SW, LW, etc...) this can really help the readers of the paper.

2) Why did you restrict the simulation in UV to a spectral range >350 nm? Especially for scattering, the UVB (290 – 315 nm) and UVA (315 – 400 nm) are very interesting, and a non-negligible part of the radiation reaching the earth and absorbed in the atmosphere is part of this spectral domain. It is a pity not to consider the 290 – 350 nm band.

3) It is a bit difficult to isolate the ocean impact and the desert impact on the radiation due to the only albedo of the ocean and of the desert with this study. The reason of this is that in this study, you consider another aerosol mixture (and extinction profile) over the ocean than the one over the desert: The aerosol mixture (and extinction profile) you consider over the ocean, is an older aerosol mixture, with less large particles. It should be valuable to make a second set of simulations over the ocean with the same aerosol mixture (and extinction profile) as the one you used for the simulation over the desert.

4) At one point (during the presentation of the database or later during the analysis of the simulation results) you need to write something about the relative quality of the three RI spectral databases (Balkanski et al., OPAC and WMO): Which one is the more modern one? Which is the more realistic one? Which one suits better to which aerosol mixture (over desert and over ocean) and why? This commented analysis would be very welcome, since the results show that for some cases and situations, the choice of the RI database is a crucial source of differences on the radiative fluxes and on the radiative forcing.

-> I definitively argue that this paper has to be accepted with only minor corrections, but if the editor decides to force you to resubmit or to make major changes, upgrading the simulations taking point 2) into account and adding a set of simulations taking point 3) into account would be a real quality gain.

**Minor comments/questions:**

*1. Introduction*
L101: "account" and "RI" written twice

*2. Data and method*
L111-112 Since we are at the beginning of a new part, please detail in the text the acronyms "DRE", "SD" and "RI"

L145 and Figure 2: Please explain very clearly about the aerosol mixture over ocean and over desert. There are different aerosol mixtures, this is clear. But do you consider a "Lagrange approach" = these are aerosol of the same plume, that is consider at a later timestamp over the ocean, or is it an "Euler approach" = you look at the mixtures at the same moment and the ocean mixture shown at this moment was former over the desert with probably at this time the same properties as the desert mixtures?

L155-159 and Figure 3: Here you can make the comments/analysis that I suggest in my comment number 4 concerning the differences of quality between the three RI datasets.

L170-171 and L175 (Figure 4): Explain better what is the aspect ratio. Is it something with the axis ratio of the ellipses of the ellipsoid? And explain the figure: If aspect ratio = 1 it is a sphere? And which aspect ratio did you use for the impact of shape further in the manuscript? The denomination "aspect ratio" will not be used anymore in the rest of the paper, therefore we do not understand why you show this graphic.

L189: simulations are done on the spectral range 0.35 – 40 micrometres. At least explain why you do not consider the main part of UV spectral range (290 – 400 nm) -> See my major comment/question number 2)

L233: You mention "the effect of the shape in the optical properties of dust" -> is the shape quantify with the "aspect ratio" mentioned in L170-171 and in Figure 4?

Table 1:
- Why did you split SW / LW at 2,5 micrometres? A rational border value is 3,5 micrometres because below 3,5 micrometres there are still solar radiation and only negligible atmospheric (thermal) emission of radiation.
- Do you have some values of the albedo (SW broadband, LW broadband, ore some values at given wavelength: 500 nm and 10 micrometres for instance)?

L275: TCWV = 10 mm over the desert: Isn't it too much? I would never expect ore than 5 mm over the desert

L289-290: A graphic with the vertical distribution of the extinction profile over ocean and atmosphere you used would be welcome

*3. Results*
L324-331: In the analysis of the results shown in Figure 9, maybe you should in the discussion compare the differences between the results to the noise: A trend seen that is below the noise should be consider with caution.

L369 (and Figure 10): "the smallest absolute differences were found for the RI used in OPAC" -> Can you explain why OPAC leads to such different results than the results obtain with the other databases. Same question for Balkanski on graphic d (TOA desert) between 500 and 2000 nm?
-> Here also it would be a good moment to discuss the what I asked in Point 4 of the major comments/questions above concerning the differences of quality between the three RI datasets.

L439: "For more realistic aerosol properties the DRE is less negative over the desert (by up to 25% for the RI of Balkanski et al. …" -> Should we understand that Balkanski is the most realistic RI description? If yes explain why.

L469: Here maybe also the real place to make the comments concerning the quality of RI database (major comments/questions point 4) above)

---

## Author Comment (AC1)

**Reviewer #3**

We acknowledge anonymous reviewer#1 for his/her constructive comments that helped us to improve the manuscript. Replies to the reviewer's specific comments are provided in the following (text in blue) after the reviewer's comments (text in black). Line numbers correspond to the manuscript version with tracked changes.

Before replying to the reviewer's comments, we must make a clarification for the used RIs.

The refractive indices that have been used for the analysis in the SW are those by Balkanski et al. (2007), Colarco et al. (2014), and OPAC, and not Balkanski et al. (2007), WMO (1983), and OPAC, as stated in the original version of the manuscript. For the LW, we used the refractive index by WMO (1983) (instead of Colarco et al. (2014)) because Colarco et al. (2014) do not provide the refractive index in the LW. We corrected this mistake and clarified what has been done in Section 2.1 (line 164).

This paper presents a sensitivity study on radiative effects due to Saharan dust aerosols. The parameter tested in the sensitivity study are 1) optical properties of aerosols under three different spectral-resolved refractive indexes (WMO 2023, OPAC: Köpke et al. 1998, 2015 and Balkanski et al. 2007). 2) Size range of aerosol size are: 0.1 - 10 μm (as done in most of former and current models) and 0.1 - 50 μm. 3) Shape form (spherical or spheroidal shapes). Calculations are made over the desert and over the ocean. The sensitivity is first led on macroscopic aerosol radiation properties (extinction, scattering and absorption coefficients), then thanks to simulations with the MYSTIC radiative transfer model (Mayer, 2009) on radiative fluxes, and at the end, the sensitivity on radiative forcing is computed and analysed.

The paper has to be understood and evaluated as an advance for radiative transfer simulations in atmospheres containing dust aerosols but not about generalities concerning dust aerosol DRE, in contrary to the wrong understanding of anonymous author of RC2 comment. In this way, the paper answers and quantifies some open questions of radiative transfer in atmosphere containing dust aerosols, a domain that needs such kind of detailed and structured studies, since the difficulty of parametrizing the radiative transfer equation in radiative transfer models for the case of dust aerosol is contained in the lack of order of magnitude and on the unknowledge about the influence of the different parameters (especially RI, size distribution and shape). Therefore, this study is a significant advance in this topic thanks to the clarifications and the quantifications that it brings. Thus, this paper is worth to be published in Atmospheric Chemistry and Physics. The paper is welled structured, the radiative transfer simulations selected are relevant, the method and the results are well explained and well presented. The paper itself is well presented, and very clear, as well from a didactic as from a linguistic aspect. Figures and tables, are presented in a clear and ergonomic way, and the results of the simulations shown in the figures are analysed in the text of the manuscript in a meticulous way.

For all these reasons, I kindly recommend to accept this article in Atmospheric Chemistry and Physics, after the minor corrections I suggest, and after the authors briefly answer the few questions here below. If the paper will be (as I suggest it) accepted with minor corrections, please give some explanations in answers to my comments

Major comments/questions

1) Please add an acronym table to define and summarize in the same place the main used acronyms (RI, TOA, BOA, AOD, DRE, IRE, SRE, SW, LW, etc...) this can really help the readers of the paper.

Table A1 has been added at the Appendix.

2) Why did you restrict the simulation in UV to a spectral range >350 nm? Especially for scattering, the UVB (290 – 315 nm) and UVA (315 – 400 nm) are very interesting, and a nonnegligible part of the radiation reaching the earth and absorbed in the atmosphere is part of this spectral domain. It is a pity not to consider the 290 – 350 nm band.

We agree with the reviewer that it would be interesting to extend the simulations to the region 290 – 350 nm and see how different dust parameterizations affect the modeling of the UV radiation. However, we did not do that for the following reasons:

1) In order to include the large spheroidal particles (i.e., radius up to 50 μm) in our calculations, we had to limit the minimum wavelength to the value of 350nm (the scattering calculations provided for spheroidal particles by MOPSMAP are for a maximum size parameter of 1005)
2) The refractive indices are more uncertain at such short wavelengths.
3) Interactions with ozone (especially tropospheric) would further increase uncertainties.
4) As can be perceived by Figure 10, not including wavelengths below 350 nm has a minor impact on the calculation of the integrals of the differences.

3) It is a bit difficult to isolate the ocean impact and the desert impact on the radiation due to the only albedo of the ocean and of the desert with this study. The reason of this is that in this study, you consider another aerosol mixture (and extinction profile) over the ocean than the one over the desert: The aerosol mixture (and extinction profile) you consider over the ocean, is an older aerosol mixture, with less large particles. It should be valuable to make a second set of simulations over the ocean with the same aerosol mixture (and extinction profile) as the one you used for the simulation over the desert.

As recommended by the reviewer, a second set of simulations was performed for the ocean using the same aerosol mixture as over the desert. In the revised version, the relevant information has been added in section 3.2.2 of the manuscript. New graphs have been also added in the supplement (Figs A5 and A6).

The new text is the following:

Since different albedo and temperature of the two surfaces, desert and ocean, also impact the differences between the simulations for SD10 and SD50 we tried to quantify at what extent they are responsible for the results shown in Figs 11 and 12. To estimate the impact of including large particles in the simulations with respect to surface albedo and surface temperature we repeated the simulations using the Balkanski et al. (2007) RI and the same SD, aerosol vertical profiles, and TCWV over the ocean and the desert (for the surface albedo and temperatures that have been assumed for the two surface types). The results are shown in Figs A5 and A6 in the supplement for the BOA and the TOA respectively. The results change by less than 1% and 2% over the BOA and the TOA respectively for both the SW and the LW, which shows the minor role of surface temperature and surface albedo for the differences over the desert and the ocean in Figs 11 and 12.

And the figures:

[Figure]

Figure A5: Relative % differences between the irradiances in the SW (blue) and LW (red) at BOA, considering SD10 and SD50 dust particles, with spheroidal shapes (FSD50,spheroids-FSD10,spheroids, Eq. 4), for the same aerosol SD and vertical distribution and the same atmospheric parameters. The differences are presented for calculations above desert and ocean for the Balkanski RI, and for different DODs and SZAs of the dust particles.

[Figure]

**Figure A6:** Similar to Fig. A5 for TOA.

4) At one point (during the presentation of the database or later during the analysis of the simulation results) you need to write something about the relative quality of the three RI spectral databases (Balkanski et al., OPAC and WMO): Which one is the more modern one? Which is the more realistic one? Which one suits better to which aerosol mixture (over desert and over ocean) and why? This commented analysis would be very welcome, since the results show that for some cases and situations, the choice of the RI database is a crucial source of differences on the radiative fluxes and on the radiative forcing.

We agree with the reviewer that the provision of such information is useful and improves the manuscript. Thus we have added relevant information (lines 167 – 169 in Section 2):

"The lower imaginary part of the RI by Balkansky et al., (2007) is considered to be more appropriate for accurately representing dust properties over the region of the campaign (Rocha-Lima et al., 2018; Ryder et al., 2019) relative to the OPAC and Colarco et al., (2014) RIs."

I definitively argue that this paper has to be accepted with only minor corrections, but if the editor decides to force you to resubmit or to make major changes, upgrading the simulations taking point 2) into account and adding a set of simulations taking point 3) into account would be a real quality gain.

Minor comments/questions:

1. Introduction

L101: "account" and "RI" written twice

corrected

2. Data and method

L111-112 Since we are at the beginning of a new part, please detail in the text the acronyms "DRE", "SD" and "RI"

Done

L145 and Figure 2: Please explain very clearly about the aerosol mixture over ocean and over desert. There are different aerosol mixtures, this is clear. But do you consider a "Lagrange approach" = these are aerosol of the same plume, that is consider at a later timestamp over the ocean, or is it an "Euler approach" = you look at the mixtures at the same moment and the ocean mixture shown at this moment was former over the desert with probably at this time the same properties as the desert mixtures?

We do not consider either, since the observations we used are derived as a mean of the observations conducted at different instances/days above ocean and desert during the Fennec campaign, (a detailed presentation about the in-situ measurements can be found in Ryder et al. (2013a), (2013b) and (2019)).

L155-159 and Figure 3: Here you can make the comments/analysis that I suggest in my comment number 4 concerning the differences of quality between the three RI datasets.

Some discussion has been added as recommended by the reviewer (see reply to major comment #4).

L170-171 and L175 (Figure 4): Explain better what is the aspect ratio. Is it something with the axis ratio of the ellipses of the ellipsoid? And explain the figure: If aspect ratio = 1 it is a sphere? And which aspect ratio did you use for the impact of shape further in the manuscript? The denomination "aspect ratio" will not be used anymore in the rest of the paper, therefore we do not understand why you show this graphic.

The aspect ratio of the spheroidal shape we consider as the dust particle shape is the ratio of the longest to the shortest diameter of the corresponding spheroid. If the aspect ratio is 1, then the particle is a sphere. In our work we consider that the dust particles have different spheroidal shapes described with the aspect ratio distribution shown in Fig. 4. The same explanation has been added to the manuscript.

L189: simulations are done on the spectral range 0.35 – 40 micrometres. At least explain why you do not consider the main part of UV spectral range (290 – 400 nm) -> See my major comment/question number 2)

See the reply to major comment #2

L233: You mention "the effect of the shape in the optical properties of dust" -> is the shape quantify with the "aspect ratio" mentioned in L170-171 and in Figure 4?

Yes, it is. See our reply in previous comment.

Table 1:

- Why did you split SW / LW at 2,5 micrometres? A rational border value is 3,5 micrometres because below 3,5 micrometres there are still solar radiation and only negligible atmospheric (thermal) emission of radiation.

Indeed, it is more common to set the border to longer wavelengths (e.g., 3.5 μm). However, at wavelengths 2.5 – 3.5 μm nearly all solar radiation is absorbed by water vapor in the atmosphere, and thus any changes in aerosol properties play a negligible role in this region. Thus, we preferred to take into account possible changes in the LW due to atmospheric emission.

- Do you have some values of the albedo (SW broadband, LW broadband, or some values at given wavelength: 500 nm and 10 micrometres for instance)?

Relevant information has been added in Section 2.3 (lines 306 - 311), as well as in Figure A2 in the appendix:

"Based on the extinction coefficient profiles and the assumed dust optical properties, the mass concentration profiles have been estimated and used as inputs for the simulations over the desert and the ocean (Fig. A1 in the appendix). It should be mentioned however that, at least for the SW, the aerosol profile has minor impact on the amount of radiation that finally reaches the top or the bottom of the atmosphere (Fountoulakis et al., 2022). The surface albedo used for the simulations in the LW is ~0.05 at 10 μm. For the SW it depends more significantly on the type of the surface (Fig. A2)"

[Figure]

Figure A2: Surface albedo that has been used for the simulations over the desert and the ocean (International Geosphere Biosphere Programme (IGBP); Loveland and Belward, 1997).

L275: TCWV = 10 mm over the desert: Isn't it too much? I would never expect ore than 5 mm over the desert

Analysis of TCWV from MERRA-2, as well as from MODIS-Aqua shows that TCWV over western Sahara ranges from ~ 5 mm to ~25 mm depending on season. Thus, TCWV can be even higher than 10 mm, especially in the summer months.

Different studies report different values of the average relative humidity over different regions of the Sahara desert. For example, Zaiani et al. report an average value of ~ 8 mm at Tamanraset in June (month of the campaign). Even larger numbers are reported in other studies (e.g., Schrijver et al.)

Zaiani, M.; Irbah, A.; Djafer, D.; Listowski, C.; Delanoe, J.; Kaskaoutis, D.; Boualit, S.B.; Chouireb, F.; Mimouni, M. Study of Atmospheric Turbidity in a Northern Tropical Region Using Models and Measurements of Global Solar Radiation. Remote Sens. 2021, 13, 2271. https://doi.org/10.3390/rs13122271

Schrijver, H., Gloudemans, A. M. S., Frankenberg, C., and Aben, I.: Water vapour total columns from SCIAMACHY spectra in the 2.36 μm window, Atmos. Meas. Tech., 2, 561–571, https://doi.org/10.5194/amt-2-561-2009, 2009.

L289-290: A graphic with the vertical distribution of the extinction profile over ocean and atmosphere you used would be welcome

Some relevant discussion has been added in Section 2.3. A graph showing the vertical distribution of the mass concentration has been added in the appendix (Figure A2).

[Figure]

**Figure A1:** Dust concentration that has been assumed for the libRadtran simulations over the desert and the ocean.

3. Results

L324-331: In the analysis of the results shown in Figure 9, maybe you should in the discussion compare the differences between the results to the noise: A trend seen that is below the noise should be consider with caution.

Relevant discussion has been added in the manuscript (lines 354 - 356):

The differences that are shown in Fig 8 (for BOA) are comparable to the level of statistical noise and should be treated with caution. While the absolute magnitude of the differences for TOA shown in Fig 9 should be also treated with caution, the change in the sign of the differences is possibly real, and not the result of statistical noise.

L369 (and Figure 10): "the smallest absolute differences were found for the RI used in OPAC" -> Can you explain why OPAC leads to such different results than the results obtain with the other databases. Same question for Balkanski on graphic d (TOA desert) between 500 and 2000 nm?

The following text has been added in the document (lines 398 - 404):

The main reason why the smallest absolute differences in UV-VIS were found for OPAC is possibly that in this spectral region differences between the SSA for SD10 and SD50 are smaller relative to the corresponding differences for Balkanski et al. (2007) and WMO RIs (Figure 6g), which means that absorption of the solar radiation at this wavelengths changes less (relative to Balkanski et al. (2007) and WMO) with the inclusion of large particles in the dust mixture. At the NIR the differences in SSA for SD10 and SD50 are much smaller for Balkansky et al. (2007) relative to the other two RIs, which correspondingly results in smaller differences between the irradiances (Figure 10)

-> Here also it would be a good moment to discuss the what I asked in Point 4 of the major comments/questions above concerning the differences of quality between the three RI datasets.

We already did that earlier in the manuscript (see reply to comment 4)

L439: "For more realistic aerosol properties the DRE is less negative over the desert (by up to 25% for the RI of Balkanski et al. …" -> Should we understand that Balkanski is the most realistic RI description? If yes explain why.

Although The RI by Balkanski et al., (2007) is indeed the most realistic for the region of study, as it is now clarified in Section 2 of the manuscript, here when we write "more realistic dust properties" we mean that we are using SD50 and spheroids instead of SD10 and spheres. We changed the manuscript in order to make it clearer.

L469: Here maybe also the real place to make the comments concerning the quality of RI database (major comments/questions point 4) above)

We already did that earlier in the manuscript (see reply to comment 4)

---

## Author Comment (AC2)

**Reviewer #1**

We acknowledge anonymous reviewer#1 for his/her constructive comments. Replies to the reviewer's specific comments are provided in the following (text in blue) after the reviewer's comments (text in black). Line numbers correspond to the manuscript version with tracked changes.

Before replying to the reviewer's comments, we must make a clarification for the used RIs.

The refractive indices that have been used for the analysis in the SW are those by Balkanski et al. (2007), Colarco et al. (2014), and OPAC, and not Balkanski et al. (2007), WMO (1983), and OPAC, as stated in the original version of the manuscript. For the LW, we used the refractive index by WMO (1983) (instead of Colarco et al. (2014)) because Colarco et al. (2014) do not provide the refractive index in the LW. We corrected this mistake and clarified what has been done in Section 2.1 (line 164).

This paper presents optical properties of aerosols under three different spectral-resolved refractive indexes. Two ranges for the aerosol size are used, below 10 μm and below 50 μm. Calculations are made over the desert and over the ocean. Moreover, the shape is considered since calculations are made with spherical and spheroidal shapes. The first part of the manuscript presents the spectral dependence of the extinction, scattering and absorption coefficients. The second part refers to a radiative transfer model and irradiances are calculated for specific situations. The authors emphasise the influence of the particle size and shape on the calculated properties.

The paper is quite detailed and complete since an extensive work has been made. Consequently, it could be published in Atmospheric Chemistry and Physics after the introduction of the following minor changes.

The paper structure should be more detailed at the introduction end. Moreover, since varied situations and conditions are considered, the authors should highlight the most relevant ones. For instance, the authors could select the noticeable results that may be followed in further research. Finally, the authors should note the restrictions of their research.

More discussion (lines 86 - 91) and additional (recent) references have been added in the introduction of the document. Furthermore, in the Summary and Conclusions sections we added more information to point out the weaknesses of our study. In particular:

The first paragraph of the "Summary and Conclusions" section has been modified as follows:

"Our study focuses on quantifying the radiative effect of the underestimation of the size of dust particles and the misrepresentation of their RI and shape in models under different atmospheric and land surface conditions. It must be clear that it is not providing quantitative estimates of the dust radiative effects on a regional or a global scale. Although our findings are not directly comparable with regional or global average DREs, they are meaningful for comparison with actual measurements at related experiments (see e.g. see Otto et al., 2007). As can be also perceived by the findings of other recent studies (e.g., (Li et al., 2022, 2021; Song et al., 2022) the estimates of DRE on larger scales are strongly affected by the inhomogeneities in the characteristics of dust depending on its sources, on changes in its chemical and physical properties as it is transferred, and on the surface and environmental conditions. The present study contributes towards understanding the uncertainties in the estimates of regionally/globally averaged DREs by models, that are commonly based on many assumptions (regarding e.g., the shape, size, and composition of dust). Isolating the effects in a microscale sensitivity study can indirectly help modelers towards understanding the importance of each factor and their combined effects, on determining the input uncertainties and their propagation to the outputs. In addition to the assessment of the regional/global DRE and its modelling

parameterization, studies that are similar to the present are also useful for example, for evaluating satellite-based data or radiative closure studies."

We have also added the following information in lines 649 – 652: "It must be also noticed that the dust shapes are various, and usually irregular, and a single model (e.g., spheres or spheroids) cannot represent accurately the complex shapes of dust (Luo et al., 2022; Connolly et al., 2020; Kalashnikova and Sokolik, 2002). Thus, further research is necessary in order to determine more precisely the effect of shape in RT modelling."

Minor remarks.

1. 88. Revise "aa model".

done

2. 100. One "considered" must be supressed.

done

3. 101. One account must be supressed, "accountthe" must be "account the", and "RIRI" must be "RI".

done

References should follow the journal style. Some of them are quite old.

References have been updated throughout the document.

---

## Author Comment (AC3)

**Reviewer #2**

We acknowledge anonymous reviewer#2 for his/her comments that helped us to improve the manuscript. Replies to the reviewer's specific comments are provided in the following (text in blue) after the reviewer's comments (text in black). Line numbers correspond to the manuscript version with tracked changes.

Before replying to the reviewer's comments, we must make a clarification for the used RIs.

The refractive indices that have been used for the analysis in the SW are those by Balkanski et al. (2007), Colarco et al. (2014), and OPAC, and not Balkanski et al. (2007), WMO (1983), and OPAC, as stated in the original version of the manuscript. For the LW, we used the refractive index by WMO (1983) (instead of Colarco et al. (2014)) because Colarco et al. (2014) do not provide the refractive index in the LW. We corrected this mistake and clarified what has been done in Section 2.1 (line 164).

This paper documents a sensitivity study to investigate and compare the sources of uncertainty in the estimate of dust direct radiative effects (DRE). Three major sources of uncertainty are considered, dust spectral refractive index (RI), dust particle size and dust particle shape. The magnitude of each uncertainty source is estimated from the differences of DRE estimated based on different assumptions of RT, size and shape. For example, the uncertainty due to dust RI is estimated by computing and contrasting the DRE of dust based on three sets of RI data with different absorptions. The sensitivity studies are performed for different surface conditions (desert vs. ocean) and solar zenith angles. The results from the sensitivity study suggests that the leading uncertainty in dust DRE computation is the dust RI, followed by dust size. The shape of dust has only negligible effects on dust DRE.

I have several main concerns and reservations about this study, which are summarized below. As a result, I have a hard time seeing how this paper advances our understanding of dust DRE and therefore do not think the paper should be accepted for publication in ACP.

My first main concern is that the sensitivity studies seem too simple to capture the variability of dust aerosols in reality, and the complex environments in which the dust aerosols are found. The objective of this study is to understand the uncertainty in dust DRE (i.e., the term). In my view, this objective can only be achieved after a reasonable estimation of the mean DRE (i.e., the term). In this study, the DRE computations are performed in a quite simple and idealized set up. For example, only two types of surfaces (desert vs. ocean) and three values of solar zenith angles are considered. The surface temperature and atmospheric profiles are assumed to be constant without considering the diurnal cycles which can be quite strong in the desert regions. As a result of this rudimentary case set up, the meaningfulness of the mean DRE and therefore DRE uncertainty is quite questionable.

The paper presents the results of a sensitivity study of dust aerosol effects on solar radiation. The main idea comes exactly from reading the suggested papers. Can someone isolate size, shape, and RI effects as a function of various AODs, solar elevations and albedo? To do so we should not deal with "mean dust DRE" which in such a sensitivity study has minimal meaning exactly because diurnal, seasonal and spatiotemporal changes in AODs, shape, size, temperature/albedo, even solar elevation ranges in a season/location have been mixed in order to define this mean DRE. Which is of course (Dust mean DRE) the outcome (the number) that the scientific community would like to have in the end, in order to assess dust effects on climate.

As the reviewer surely understands, the domain is complex and structured studies could be useful for the community to quantify individual effects and to understand their complex response to realistic changes of basic atmospheric and solar parameters.

So, it is probably not possible to use our results for an overall assessment of the e.g. Saharan dust effect on Earth – atmosphere balance, but it can help scientists dealing with measurements and models in related experiments, to focus on the most uncertain aspects of dust parameterizations needed for radiative forcing studies. And most probably, eventually, to indirectly help improving global or regional models too.

The conditions are not realistic for using them to directly retrieve a (regional or global) mean DRE, however they are meaningful when compared with actual measurements at related experiments (e.g. see Otto et al., 2007). Such studies are not dealing only with a mean DRE assessment and its modelling parameterization but also with other aspects. E.g., evaluating satellite-based data that need ground-based measurements and detailed dust optical properties for a certain location and a certain solar elevation in order to be meaningful for validation or radiative closure studies.

We have tried to make all the above clearer, especially in the introduction and the conclusion sections (see e.g., lines 600-613 at the summary and conclusions section).

Finally, a possible answer to this comment has been provided also from rev 3 that we point again here :

"The paper has to be understood and evaluated as an advance for radiative transfer simulations in atmospheres containing dust aerosols but not about generalities concerning dust aerosol DRE, in contrary to the wrong understanding of anonymous author of RC2 comment. In this way, the paper answers and quantifies some open questions of radiative transfer in atmosphere containing dust aerosols, a domain that needs such kind of detailed and structured studies, since the difficulty of parametrizing the radiative transfer equation in radiative transfer models for the case of dust aerosol is contained in the lack of order of magnitude and on the unknowledge about the influence of the different parameters (especially RI, size distribution and shape). Therefore, this study is a significant advance in this topic thanks to the clarifications and the quantifications that it brings."

For example, one of the main conclusions is that "At the top of the atmosphere (TOA) close to dust sources, the underestimation of size issues an underestimation of the direct warming effect of dust of ~18 - 25 W/m2, for dust aerosol optical depth (DOD) of 1 at 0.5 μm". What is the meaningfulness of the DRE 18 - 25 W/m2? Is it a regional mean value? Is it instantaneous value or diurnally averaged? How could a modeler compare their DRE simulations to such DRE values reported in this study?

We agree that most of the studies targeting a regional mean DRE value, use a certain DRE scale that would have a meaning for more global radiative forcing assessment. So, in our case percentage changes have more meaning. However, absolute solar radiation changes in a focused radiative transfer modeling sensitivity study is exactly what it is defined. It is the difference in W/m$^2$ among two defined runs. Which is not, and it does not have to be, comparable with model simulations of a modeler interested in a mean dust DRE. However, maybe a modeler could think that for example, diurnal or seasonal patterns of e.g. AOD, or size through transport, or RI, would affect the mean DRE due to the different response to solar elevation but that will not happen based on the size. And then prioritize the efforts towards dealing with such issues. So, someone can improve more his/her models towards less uncertain results linked with spatiotemporal assumptions of dust properties and their variability.

As a suggestion for revision (should the authors consider resubmission), I would recommend the study to set up the case studies in a more realistic and meaningful context that hopefully can be compared to the model simulations or other observations studies quantitatively.

As it is already stated in the original version of the manuscript (lines 111 - 114) and explained above: "It is important to clarify at this point that our study aims to contribute towards the understanding of how the

dust optical properties parameterization in models affect the calculated dust DRE under different atmospheric and land surface conditions, rather than providing quantitative estimates of the (local or global) dust radiative effects."

We however tried to follow the suggestion and address this comment by further analyzing the sensitivity of our results to surface temperature over the desert, as well as the effect of surface albedo (by performing simulations for the same size distribution over the two surfaces). The results are discussed in Section 3.2.2 while two new graphs have been added in the Appendix (Figures A3 and A4)

In comparison with several previous modeling studies (e.g., a recent one by Li et al. 2022) or the observations studies (e.g., Song et al. 2022), this study seems to be rudimentary and lack novelty. As mentioned above, the setup of the case studies is too simple. Moreover, the main conclusion from this study seems to be identical to previous ones (e.g., Li et al. 2021, 2022, Song et al. 2022). So, the novelty and significance of this study are quite questionable. They should be clearly explained and justified.

Indeed, there are other recently published studies that deal with the same problems as sensitivity studies without mentioning mean DRE regional assessments. It is true that especially the Song et al. study, which presents similar findings to our study, appeared at the time of the submission of this work. This led us to the inclusion of spectral irradiance aspects, focus more on the physical mechanisms that control the radiative transfer processes and also create section 4 that exactly deals with this aspect comparing our findings with the findings of other studies pursuing similar aspects.

One of the new aspects included in this study is the spectral simulations that, as discussed in Otto et al., (2007), are very useful for experimental campaigns where spectral dependence of different aerosol properties could play a significant role relative to the propagation of the spectral and total solar radiation through the dust layer.

E.g. what is the mean spectral lidar ratio variability in an area as large as the Sahara that must be used in order to include space lidars in model inputs aiming for a mean dust DRE? And what is the spectral AOD dependence effect on SW radiation in this whole area when the only source of information is satellite-based data dealing with mainly one AOD wavelength? Such uncertainties affect significantly the DREs calculated at larger scales using such data, and studies such as the present contribute towards the better understanding of their impact.

Finally, we see our study as an independent verification of part of the Song et al., 2022 findings that have been conceived, of course with different set ups and assumptions. In addition, we tried to interpret with physical mechanisms some of the aspects concluded here.

In paragraph 4 we have added, already in the original version of the manuscript, some aspects pointing out differences and new findings of our study compared with the ones mentioned here.

Several places of the paper are confusing and/or misleading. For example, in most previous studies (e.g., Kok et al. 2017, all of Ryder et al. papers) dust particle size distribution was described in terms of dust geometrical diameter whereas this study uses radius. Although this is not physically wrong, making it difficult to compare this study with previous ones. In Fig. 13 to Fig. 16, the comparison of LW DRE with SW DRE at different SZA is confusing because the LW DRE is independent of SZA. A more meaningful comparison is the diurnally averaged DRE.

Probably the radius or diameter aspect is a detail, but we used it mostly as it is the input in the RT models. Since we are not aiming to directly compare with models dealing with mean DRE we believe that it can

reasonably stay as is. We do agree that LW DRE representation with SZA is confusing. We mentioned this in the captions and in the text but we kept it in the graphs for visual comparison with the SW in each case.

Finally, the overall quality of the paper seems to fall short of the ACP standard. The paper has many typos and many references are missing (e.g., Ryder et al. 2021, Gutleben et al., 2019; 2020, Dubovik et al., 2006 to name a few)

References:

Li, L., Mahowald, N. M., Kok, J. F., Liu, X., Wu, M., Leung, D. M., Hamilton, D. S., Emmons, L. K., Huang, Y., Sexton, N., Meng, J., and Wan, J.: Importance of different parameterization changes for the updated dust cycle modeling in the Community Atmosphere Model (version 6.1), Geosci. Model Dev., 15, 8181–8219, https://doi.org/10.5194/gmd-15-8181-2022, 2022.

Li, L., Mahowald, N. M., Miller, R. L., Pérez García-Pando, C., Klose, M., Hamilton, D. S., Gonçalves Ageitos, M., Ginoux, P., Balkanski, Y., Green, R. O., Kalashnikova, O., Kok, J. F., Obiso, V., Paynter, D., and Thompson, D. R.: Quantifying the range of the dust direct radiative effect due to source mineralogy uncertainty, Atmos. Chem. Phys., 21, 3973–4005, https://doi.org/10.5194/acp-21-3973-2021, 2021.

Kok, J. F. et al. Smaller desert dust cooling effect estimated from analysis of dust size and abundance. Nat Geosci 10, 274–278 (2017).

We tried to improve the document quality and included the suggested (as well as few more, recent) references in the introduction.